# A nuclear receptor HR96-related gene underlies large *trans*-driven differences in detoxification gene expression in a generalist herbivore

Meiyuan Ji [1,4], Marilou Vandenhole [2,4], Berdien De Beer [2,4], Sander De Rouck [2], Ernesto Villacis-Perez [2], René Feyereisen [2], Richard M. Clark [1,3] ✉ & Thomas Van Leeuwen [2] ✉

The role, magnitude, and molecular nature of *trans*-driven expression variation underlying the upregulation of detoxification genes in pesticide resistant arthropod populations has remained enigmatic. In this study, we performed expression quantitative trait locus (eQTL) mapping ($n = 458$) between a pesticide resistant and a susceptible strain of the generalist herbivore and crop pest *Tetranychus urticae*. We found that a single *trans* eQTL hotspot controlled large differences in the expression of a subset of genes in different detoxification gene families, as well as other genes associated with host plant use. As established by additional genetic approaches including RNAi gene knockdown, a duplicated gene with a nuclear hormone receptor HR96-related ligand-binding domain was identified as causal for the expression differences between strains. The presence of a large family of HR96-related genes in *T. urticae* may enable modular control of detoxification and host plant use genes, facilitating this species' known and rapid evolution to diverse pesticides and host plants.

Animals have evolved to overcome harmful compounds in their environments (xenobiotics), from microbial toxins to plant-produced specialized compounds in the case of herbivores. Additional and strong selective agents are pesticides, for which resistance evolution is common, from vectors of human and animal diseases like mosquitoes and ticks, to diverse insect and mite herbivores that impact crops[1–4]. Despite its ubiquity, and importance for human welfare in the case of pesticides, the genetic basis of xenobiotic resistance evolution is incompletely understood. Nevertheless, a major route involves changes in the metabolism, sequestration, or transport and excretion of compounds[5]. While the genetic mechanisms impacting such differences are complex, the upregulation of genes in detoxification families, including cytochrome P450 monooxygenases (CYPs), glutathione-S-transferases (GSTs), and carboxyl-choline esterases (CCEs) is well established, especially in the case of pesticide resistant strains[1,6,7].

Upregulation of detoxification genes in resistant strains has been shown to result from local, or *cis*, genetic variation (e.g., sequence changes in promoters or enhancers)[5,8–10]. A particularly well-characterized example involves the upregulation of the *Drosophila melanogaster Cyp6g1* gene associated with resistance to dichlorodiphenyltrichloroethane (DDT) and neonicotinoids[11,12]. Far less well understood is the role of *trans* genetic variation, in which genetic changes in a (usually) genetically distant factor impact the expression

[1]School of Biological Sciences, University of Utah, Salt Lake City, UT, USA. [2]Department of Plants and Crops, Faculty of Bioscience Engineering, Ghent University, Ghent, Belgium. [3]Henry Eyring Center for Cell and Genome Science, University of Utah, Salt Lake City, UT, USA. [4]These authors contributed equally: Meiyuan Ji, Marilou Vandenhole, Berdien De Beer. ✉e-mail: clark@biology.utah.edu; Thomas.VanLeeuwen@UGent.be

of a target gene (i.e., variation in components that perceive and signal exposure to xenobiotics). Variation in *trans* regulation of detoxification genes has nonetheless been suggested by genetic studies[13–18]. However, the nature of the underlying genes and allelic variation involved have remained enigmatic despite progress in the identification of regulatory pathways[19,20].

The two-spotted spider mite, *Tetranychus urticae*, is an attractive species in which to unravel the genetic underpinnings of variation impacting xenobiotic metabolism. This species, a member of the Acari within Arthropoda subphylum Chelicerata, is a generalist herbivore with an exceptionally large host range of more than 1100 plant species including many crops[21]. Mirroring its broad host range on plants that produce diverse compounds toxic to many herbivores, *T. urticae* rapidly evolves resistance to acaricides[1], which are pesticides effective against members of the Acari, and similar genetic mechanisms are likely deployed against plant defenses[22,23]. In comparisons of acaricide resistant strains to susceptible strains, heightened expression of detoxification genes has often been observed, not only for CYPs, GSTs, and CCEs, but also for genes in other families involved (or putatively involved) in metabolic resistance, such as short chain dehydrogenases (SDRs), lipocalins, ATP-binding cassette (ABC) and major facilitator superfamily (MFS) transporters[22], uridine diphosphate (UDP)-glycosyltransferases (UGTs) of bacterial origin that have greatly expanded in the *T. urticae* genome[24], and intradiol ring cleavage dioxygenases (DOGs) acquired from fungi[25]. Further, in experimental evolution studies with *T. urticae*, dramatic differences in the expression of diverse detoxification or host plant use associated genes have been reported in as few as five generations upon shift to a challenging host plant[26,27]. The scope and rapidity of these changes has raised the possibility of selection on loci that act in *trans* to concertedly impact target genes involved in xenobiotic metabolism or host plant use. Recently, we characterized allele-specific expression among a panel of *T. urticae* strains and F1 progeny, and found that *trans* variation was responsible for the high expression of numerous detoxification genes in several *T. urticae* strains with high-levels of acaricides resistance[18]. However, the experimental design did not allow us to identify specific genomic loci causal for *trans* effects, and hence the molecular nature of causal genes and pathways.

In the current study, we extended our earlier work by performing expression quantitative trait locus (eQTL) mapping between two *T. urticae* strains that vary markedly in acaricide resistance. Overall, we identified more *trans* than *cis* eQTLs. A single *trans* eQTL hotspot was associated with large differences in the expression of diverse genes associated with xenobiotic metabolism and host plant use, with the respective haplotype from the resistant strain resulting in the tens- to several hundred-fold upregulation of gene(s) in the CYP, SDR, and DOG gene families. RNA interference (RNAi) knockdown of tandemly duplicated genes located at the peak for the *trans* eQTL hotspot recapitulated most of the hotspot's impacts on detoxification gene expression. The duplicate genes, which harbor multiple coding sequence differences between the strains, encode products with ligand-binding domains (LBDs) that have homology to nuclear hormone receptors (NHRs) known to signal exposure to exogenous compounds[19,28]. Therefore, segregating genetic variation in regulators of xenobiotic pathways can be a source of the dramatic upregulation of detoxification genes associated with pesticide resistance evolution and host plant adaptation in arthropod species.

## Results

### Dense recombination in *T. urticae*
To identify sources of intra-specific variation in detoxification and host plant use associated gene expression in *T. urticae*, we first generated 100 bp paired-end RNA-seq reads from families of isogenic F3 females derived from crosses of the inbred (isogenic) strains MR-VPi and ROS-ITi (F0 generation)[18]. Strain MR-VPi, hereafter denoted as the resistant

strain, R, has a history of intense and recurrent acaricide selection, and is moderately to highly resistant to six acaricides that collectively belong to four acaricide classes with different modes of action; in contrast, strain ROS-ITi, hereafter denoted as S, is comparatively susceptible to these acaricides (Kurlovs et al.[18] and Methods). The female F3 families were constructed by crossing single F2 males produced from S × R F1 females to virgin females of strain S (458 families in total, Fig. 1a). With this design, a family of F3 females sired by a single recombinant F2 male are genetically identical full siblings, a consequence of the haplodiploid sex determination system in spider mites (unfertilized eggs develop as haploid males) and the use of inbred lines (note that for each locus in an F3 isogenic family, only two genotypes are possible, RS and SS, Fig. 1a). A median of 42 4–5-day-old F3 females were collected per F3 family to provide sufficient material for RNA extractions (female mites are only ~0.6 mm in length), as well as to lessen the impact of individual variation on gene expression phenotypes. The crosses and collection of 458 RNA samples (one RNA sample from each F3 family) were performed with mites maintained on bean leaves in the absence of acaricide selection.

Using aligned RNA-seq reads at SNP positions predicted from genomic sequencing data for the F0 generation R and S strains[18], we genotyped each F3 family, imputed genotypes, and identified a total of 2927 recombination events across the 458 families (Supplementary Data 1). A median of two recombination events were observed for each of the three *T. urticae* holocentric chromosomes (Fig. 1b). As assessed using permutations with 1.5 Mb windows, recombination events were not randomly distributed ($p < 0.001$ for each chromosome; Supplementary Fig. 1). Nevertheless, large chromosomal regions of very low recombination were not observed, and significant deviations from the expected 1:1 RS to SS genotype ratios were only observed on chromosome 2 (0–6.2 Mb; Fig. 1c, chi-square goodness of fit tests with Bonferroni correction, adjusted-*p*, or adj-*p*, < 0.01; the most significant deviation was observed at ~2.65–2.75 Mb, with an excess of RS genotypes).

### Genome-wide eQTL atlas
For genetic mapping of expression variation, we selected 1889 maximally informative genotype bins (median length of 29.5 kb) based on observed recombination events in the 458 isogenic families that constituted our eQTL mapping population (a schematic illustrating the approach to bin selection is shown in Supplementary Fig. 2; genotype bins are provided in Supplementary Data 2). Using these markers with expression phenotypes assessed with RNA-seq, we identified significant genotype-phenotype associations with a linear model (Fig. 2a; adj-*p* < 0.01; Supplementary Data 3). Of 5685 local associations, or *cis* eQTLs, which we defined as those for which the associated genotype bin midpoint was within ±800 kb of its target gene, the majority (54.7%) were within ±100 kb (Supplementary Fig. 3). In addition, we identified 10,563 distant associations, or *trans* eQTLs (Fig. 2a, b). Of the 9740 genes on chromosomes 1–3 with associated eQTLs (i.e., excluding genes on small, unplaced scaffolds), 31.6% (3082) and 24.0% (2341) of genes were regulated by single *cis* and single *trans* eQTLs, respectively, and genes with one *cis* and one *trans* eQTL accounted for an additional 17.2% (1676 genes). The remaining genes were associated with other combinations of *cis* or *trans* control (Supplementary Data 4). Of a set of 723 genes belonging to gene families associated with the metabolism (i.e., CYPs, GSTs, SDRs, or DOGs), binding (i.e., lipocalins) or transport (i.e., ABC transporters) of acaricides or plant specialized compounds (hereafter detoxification genes, Supplementary Data 5), 537 (74.3%) had at least one *cis* or *trans* eQTL. For all genes, as well as for the detoxification genes, -$\log_{10}$(adj-*p*) values for *cis* eQTL were significantly greater than for *trans* eQTL; similar trends were observed when examining absolute values of effects sizes (beta values from a linear model; Wilcoxon rank sum tests, all $p < 10^{-15}$; Supplementary Fig. 4), a finding observed in related studies in other animals and plants[29].

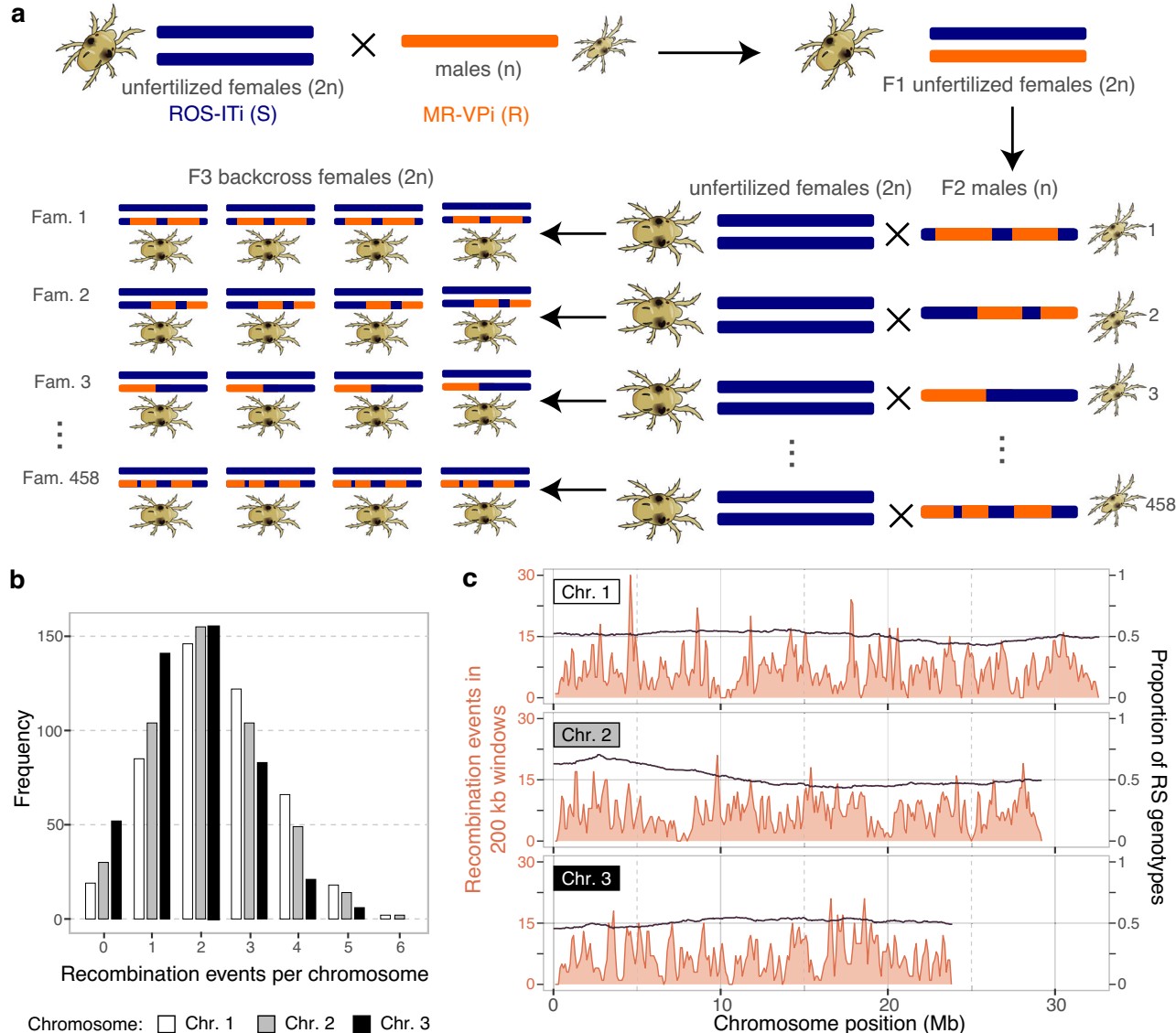

**Fig. 1 | Experimental design for eQTL mapping and distribution of recombination events. a** Diploid (2n) unfertilized females of strain ROS-ITi (susceptible strain, S, F0 generation) were collected and crossed to haploid (n) males of strain MR-VPi (resistant strain, R, F0 generation). Subsequently, haploid F2 males produced by F1 females were crossed individually to unfertilized S females (notice that the haploid F2 males have recombined chromosomes resulting from meiotic crossovers in their hybrid F1 mothers). Females in each of the resulting F3 families were full siblings and genetically identical because they each arose from a cross between one isogenic haploid male and S strain isogenic females. In total, 458 such F3 families were generated and used for RNA preparation (denoted Fam. 1–458). **b** Histogram of the recombination events for each of the three *T. urticae* chromosomes (Chr. 1–3) as assessed with the 458 F3 families using RNA-seq based genotypes. **c** The distribution of recombination events in the 458 families, as assessed with a sliding window analysis (200 kb window size with 100 kb steps; orange line; left axes), and the proportion of RS genotypes, as calculated every 50 kb (black line; right axes).

## A *trans*-eQTL hotspot controls expression of many detoxification genes

To identify loci impacting the expression of many genes (i.e., potential master regulators of gene expression), as well as the genes they regulate in *trans*, we assessed the number of *trans* eQTLs in 200 kb non-overlapping windows. We identified nine *trans*-eQTL hotspots (HS1-2, HS3-7, and HS8-9 on chromosomes 1–3, respectively) controlling the expression of ≥ 100 genes (a range of 101 for HS2 to 1125 for HS5, Fig. 2c; the coordinates for all hotspots and the number and identity of *trans*-regulated genes is provided in Supplementary Data 6, and the genes located in the hotspot intervals are given in Supplementary Data 7). For some hotspots, a bias in the number of genes up- versus downregulated by the RS genotype as compared to the SS genotype was observed (Supplementary Fig. 5; upregulation for HS1-HS3, and HS7, and downregulation for HS5-HS6, and HS8; chi-square tests with

Bonferroni correction, adj-$p < 0.05$). The hotspot impacting the most genes (HS5) was coincident with the peak of distortion in the genotype ratio toward RS at ~2.7 Mb on proximal chromosome 2 in the eQTL mapping population (Supplementary Fig. 6).

In a gene ontology (GO, based on molecular function) enrichment analysis with genes controlled in *trans* by HS1 (chromosome 1, 12.4–12.6 Mb, 182 genes), 12 terms were enriched (adj-$p < 0.05$; Supplementary Data 8), most of which were associated with acaricide or plant specialized compound metabolism or transport. These included GO:0005506 ("iron ion binding") associated with CYPs and DOGs, GO:0016758 ("transferase activity, transferring hexosyl groups") associated with UGTs, and GO:0042626 ("ATPase activity, coupled to transmembrane movement of substances") associated with ABC transporters. In total, 56 genes, pseudogenes, or gene fragments belonging to the CYP (15), GST (11), DOG (3), UGT (9), SDR (4), lipocalin

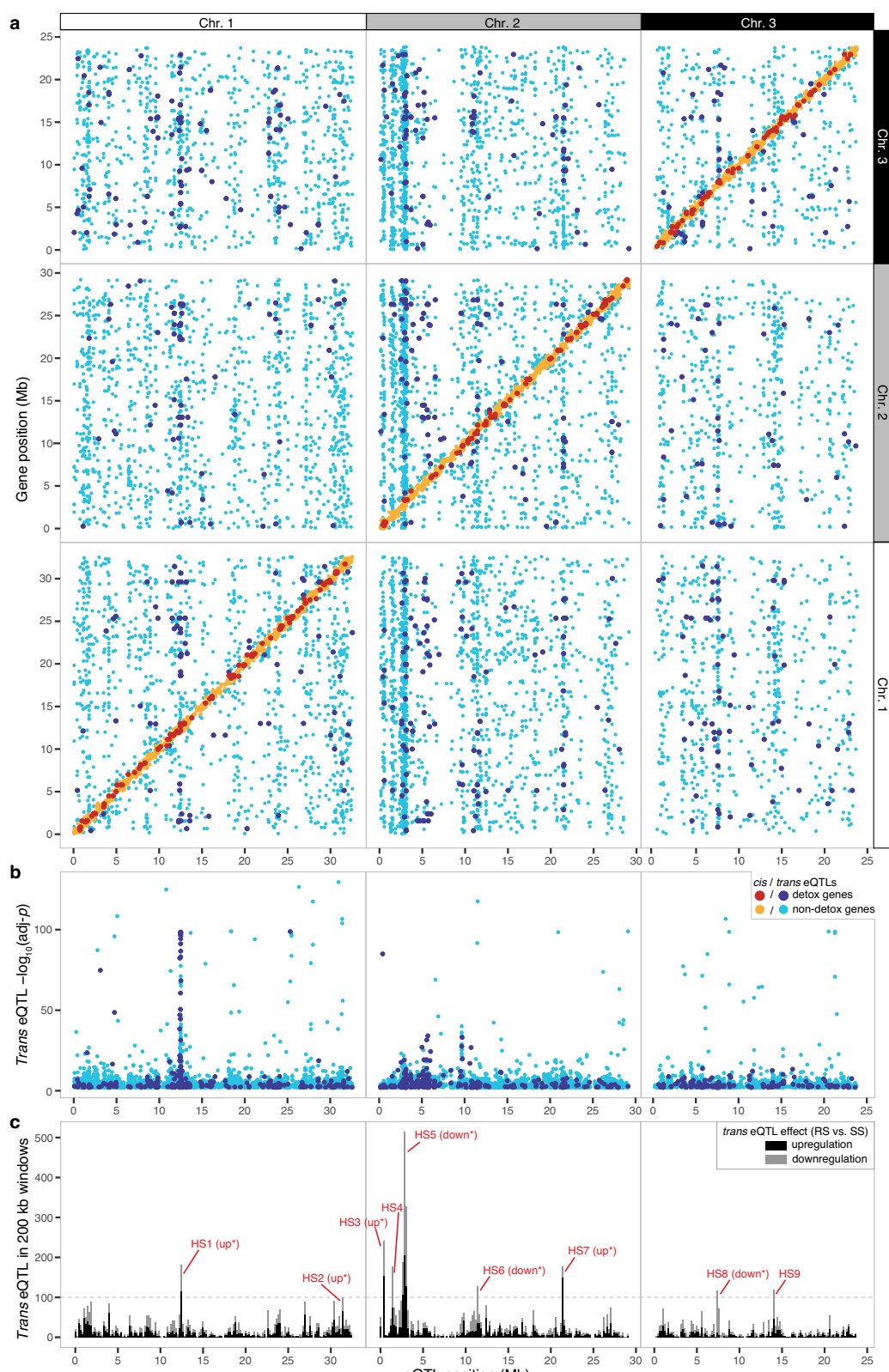

(2), CCE (2), ABC (7), and MFS transporters (3) detoxification families had significant *trans* associations with HS1. Among the CYPs with *trans*-regulation from HS1, most belonged to the CYP392 family (12 of 15), and of these 10 were upregulated by the RS genotype at HS1. These included *CYP392A11* and *CYP392A12* that were previously shown to be highly expressed in the R strain compared to several acaricide susceptible *T. urticae* strains[18]. The remaining enriched GO terms were

associated with cysteine-protease activity. Protease genes contributing to the enrichments belonged to the Cathepsin B, Cathepsin L, and legumain families, for which upregulation in *T. urticae* is likely important to overcome anti-digestive protease inhibitors produced by host plants[30].

The -log₁₀(adj-*p*) values and effect sizes for HS1 *trans* associations were on average much larger than for those observed for *trans*

**Fig. 2 | Genomic distribution of eQTLs in the R × S mapping population.**
**a** Scatter plot showing the position of significant eQTLs (x-axis, adj-$p$ < 0.01) along with the respective target genes (y-axis). Colors of points distinguish *cis* eQTLs (yellow or red) from *trans* eQTLs (light or dark blue) as indicated in the legend (bottom right); red and dark blue indicate *cis* and *trans* eQTLs for genes in the detoxification gene set, respectively; otherwise, eQTLs are associated with non-detoxification genes. For display, associations are grouped by each of the three *T. urticae* chromosomes (Chr. 1–3) as indicated at the top and far right. **b** Log transformed adj-$p$ (-log$_{10}$) values for *trans* eQTLs (y-axis) by genomic position (x-axis). Each point is for a single *trans* eQTL (light and dark blue color coding are after **a**). **c** The number of *trans* eQTLs (based on bin midpoints) for which associated target genes are either up- or downregulated (stacked black and gray bars, respectively) in non-overlapping 200 kb windows across the *T. urticae* genome (x-

axis) are indicated (legend, far upper right; up- and downregulation are defined in the comparison of expression levels of RS to SS genotypes at *trans* eQTL loci). For nine *trans* eQTL hotspots (HS1-HS9) with > 100 *trans* associations each, where a significant bias (adj-$p$ < 0.05, denoted by an asterisk) in the direction of effect conferred by *trans* regulation was observed, "up" or "down" is given in parentheses (for HS1-HS9, respectively, chi-square goodness of fit test results are: $\chi^2(1) = 12.66$, $p = 0.003$; $\chi^2(1) = 8.37$, $p = 0.035$; $\chi^2(1) = 18.00$, $p = 1.99 \times 10^{-4}$; $\chi^2(1) = 7.35$, $p = 0.060$; $\chi^2(1) = 38.09$, $p = 6.09 \times 10^{-09}$; $\chi^2(1) = 26.88$, $p = 1.94 \times 10^{-06}$; $\chi^2(1) = 85.38$, $p = 2.21 \times 10^{-19}$; $\chi^2(1) = 77.14$, $p = 1.44 \times 10^{-17}$; and $\chi^2(1) = 5.25$, $p = 0.197$; $p$ values adjusted with the Bonferroni method to account for multiple tests). For **a** and **b**, the $p$-values for eQTLs, corrected for multiple tests (adj-$p$ values), are from the output of MatrixEQTL.

associations genome-wide, or for those at the other eight hotspots (Fig. 2b, c, Supplementary Fig. 5). Nevertheless, for genes controlled in *trans* by HS6 (chromosome 2, 11.4–11.6 Mb, 172 genes) and HS7 (chromosome 2, 21.4–21.6 Mb, 183 genes), one or more GO terms associated with CYPs, cysteine proteases or DOGs were enriched, albeit with modestly significant $p$-values as compared to those for HS1. For all the remaining hotspots, except for HS2 (chromosome 1, 31.4–31.6 Mb, 102 *trans*-regulated genes) and HS3 (chromosome 2, 0.4–0.6 Mb, 242 *trans*-regulated genes), at least one GO term was enriched. These included, for example, GO:0005216 ("ion channel activity") for HS4 (chromosome 2, 1.4–1.6 Mb, 240 *trans*-regulated genes) and GO:0004930 ("G protein-coupled receptor activity") for HS5 (chromosome 2, 2.6–3.2 Mb, 1125 *trans*-regulated genes) (Supplementary Data 8).

## Characterization of HS1

Because HS1 was unique in our study in its magnitude of effect on *trans*-regulation of detoxification genes, we focused our subsequent efforts on validating and characterizing this hotspot. To do this, we first constructed two independent sets of near-isogenic lines (NILs) by marker-assisted backcrossing in which the R haplotype at HS1 was introgressed into the S genetic background (see Methods and Supplementary Fig. 7). For each set, one NIL was homozygous for the R haplotype in a small interval ( < 0.8 Mb) at HS1 (A-NIL-HS1$^{RR}$ and B-NIL-HS1$^{RR}$), and the other (control) NIL was homozygous for the S haplotype at HS1 (A-NIL-HS1$^{SS}$ and B-NIL-HS1$^{SS}$, respectively). The R strain introgressions into A-NIL-HS1$^{RR}$ and B-NIL-HS1$^{RR}$ harbored all or the majority of the genotypic bins in the 200 kb window for HS1 (Fig. 3a and Supplementary Data 9). As revealed by differential gene expression analyses with RNA-seq data from adult females in comparisons of A-NIL-HS1$^{RR}$ to A-NIL-HS1$^{SS}$ and of B-NIL-HS1$^{RR}$ to B-NIL-HS1$^{SS}$, many differentially expressed genes (DEGs; adj-$p$ < 0.01, absolute log$_2$ fold change, or log$_2$FC, > 0.5, see Methods) controlled in *trans* were shared (Fig. 3b; DEGs within the introgressed regions, and that are likely due primarily to *cis* effects, are not shown, Supplementary Data 10 and Supplementary Data 11). Additionally, DEGs for both sets of NILs were highly over-represented in the set of genes with *trans* associations to HS1 identified by eQTL mapping (Fig. 3b, one-tailed hypergeometric tests: A-NIL-HS1 vs. eQTLs HS1, $p = 3.76 \times 10^{-65}$; B-NIL-HS1 vs. eQTLs HS1, $p = 1.76 \times 10^{-63}$; A-NIL-HS1 vs. B-NIL-HS1, $p = 2.57 \times 10^{-64}$). Among genes with overlaps among the three comparisons, 45% belonged to detoxification families (where no overlap was observed, the respective values were ~9–22%).

We also examined the relationships between the fold changes for genes with *trans* associations to HS1 and for DEGs identified in the NIL comparisons. For the majority of genes with strong *trans* regulation by genotype bins at HS1, elevated expression was associated with the RS genotype (Fig. 3c). This was especially striking for the genotype bin centered at 12.507 Mb, which harbored the genes with the most striking upregulation, including *CYP392A11, CYP392D2, CYP392D8, DOG11*, and *UGT204B1*, as well as several CYPs annotated as

pseudogenes in the reference assembly (e.g., *CYP392D5p* and *CYP392D10p*). Strikingly, these genes, as well as many others *trans*-regulated by the 12.507 Mb genotype bin, or by the nearby bins (e.g., *CYP392EnP, CCE58, GSTd09*, and *GSTd14*), were also similarly upregulated in both of the A-NIL-HS1$^{RR}$ versus A-NIL-HS1$^{SS}$ and B-NIL-HS1$^{RR}$ versus B-NIL-HS1$^{SS}$ comparisons. In some cases, as for *CYP392D2, CYP392A11, CYP392D8*, and *DOG11*, the observed upregulation was dramatic, with log$_2$FC values as high as ~8 for *CYP392A11* in both the NIL set comparisons (Fig. 3d). Where overlap was not observed between HS1 *trans*-regulated genes and DEGs identified in the NIL comparisons, fold changes were generally small (absolute log$_2$FC < 1.0, Fig. 3c, d and Supplementary Data 11).

To understand the mode of inheritance for expression traits associated with allelic variation at HS1, we also collected matching expression data for F1 females derived from A-NIL-HS1$^{RR}$ × A-NIL-HS1$^{SS}$ and B-NIL-HS1$^{RR}$ × B-NIL-HS1$^{SS}$ crosses. For a set of genes strongly up-regulated in *trans* by the R haplotype at HS1, as well as in A-NIL-HS1$^{RR}$ and B-NIL-HS1$^{RR}$, expression levels in F1s were intermediate in most cases, revealing a dosage dependence of the *trans*-regulatory factor(s) at HS1 (Fig. 3e and Supplementary Data 10).

## An epistatic interaction underlies heightened expression of *CYP392A12* in the R strain

For 79 of the 182 (43.4%) genes with *trans* associations to HS1, a *cis* eQTL was also identified (Fig. 4a). In an examination of the 79 *cis* eQTLs, we found that the (local) RS genotype more often resulted in down- than upregulation as compared to the SS genotype (54 versus 25, chi-square goodness of fit test: $\chi^2(1) = 10.65$, $p = 0.001$; for all genes with *cis* effects, no bias was observed: $\chi^2(1) = 3.30$, $p = 0.069$). While the relevance of this observation is not clear, one of the 25 genes for which a *cis* effect was associated with upregulation by the RS genotype was *CYP392A12*. This gene is the most similar CYP to *CYP392A11* in *T. urticae*[21], and *CYP392A11* and its close homologs in *T. urticae* appear to be important in the development of acaricide resistance as demonstrated by recent genetic and functional studies[10,31]. *CYP392A12* is located 27.4 kb from *CYP392A11* on distal chromosome 1 (Supplementary Data 5), and like *CYP392A11* has a *trans* association with higher expression associated with the RS genotype at HS1. An unanticipated finding, however, was that *CYP392A12* was not a DEG in either the A-NIL-HS1$^{RR}$ versus A-NIL-HS1$^{SS}$ or the B-NIL-HS1$^{RR}$ versus B-NIL-HS1$^{SS}$ comparisons (Supplementary Data 10). Therefore, it was one of the few detoxification genes with large expression changes attributable to HS1 that was not validated in the differential gene expression analyses with the HS1 NIL sets.

To investigate this discrepancy, we assessed the expression of *CYP392A12* in the 458 F3 families of the eQTL mapping population by conditioning on both the genotype at HS1 (*trans* eQTL) and at the genotype bin for the *CYP392A12 cis* eQTL that overlaps with the gene. This analysis revealed that among the four possible genotypic combinations, high-level expression of *CYP392A12* was only observed when RS genotypes at both HS1 and the *cis* eQTL were present (all other

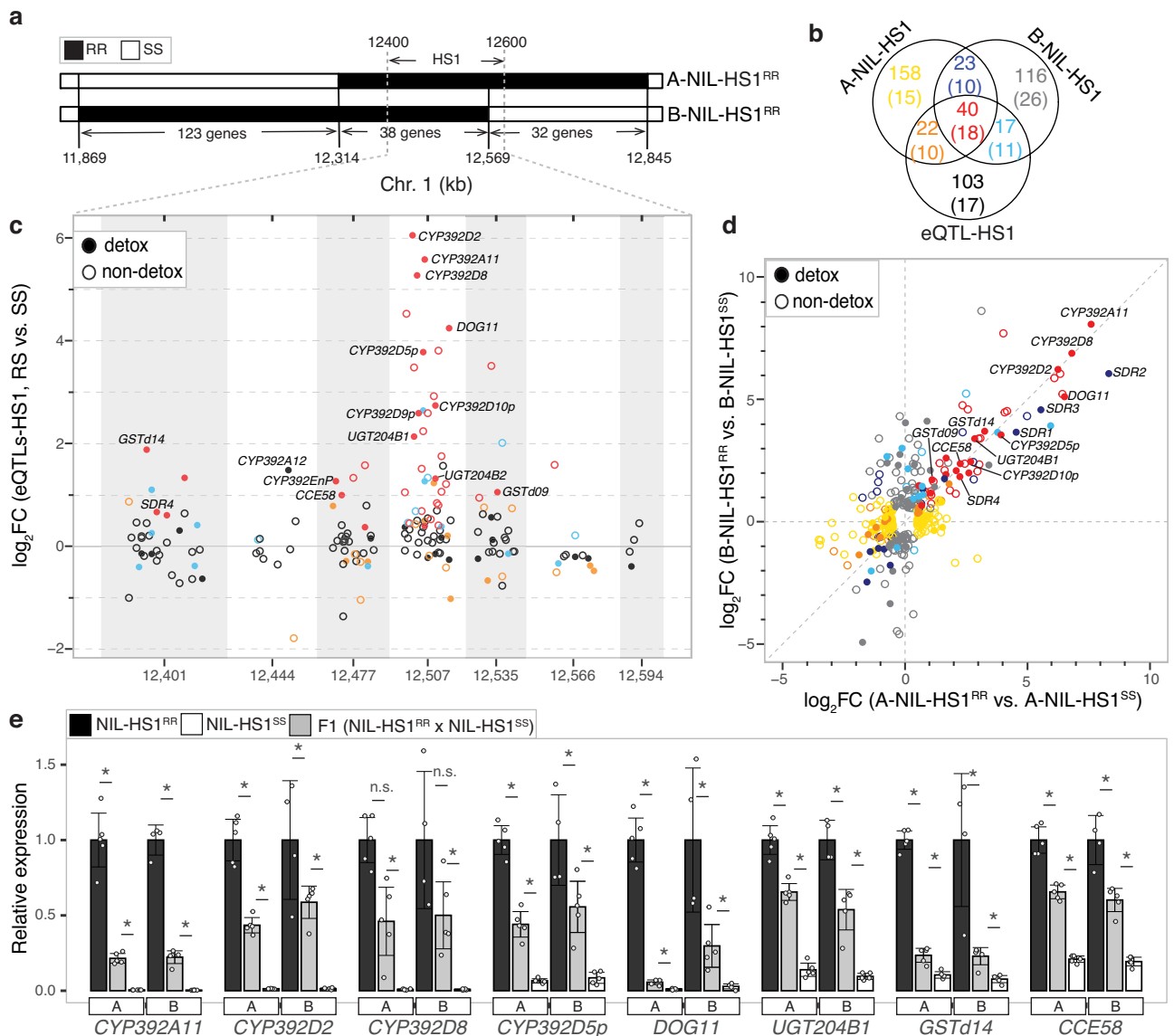

**Fig. 3 | Validation of *trans* eQTL HS1 using near-isogenic lines (NILs). a** Genomic region at and nearby the 200 kb HS1 window (12.4–12.6 Mb) on chromosome 1 showing the extent of the introgressed R haplotype in A-NIL-HS1[RR] and B-NIL-HS1[RR] (legend, upper left). The numbers of genes within the introgressed regions are indicated at the bottom. **b** Venn diagram showing the number and overlap of differentially expressed genes (adj-*p* < 0.01, and see Methods) in the comparison of A-NIL-HS1[RR] versus A-NIL-HS1[SS] (denoted A-NIL-HS1), B-NIL-HS1[RR] versus B-NIL-HS1[SS] (denoted B-NIL-HS1), and those genes with *trans* control originating from HS1 as determined by eQTL mapping (denoted eQTL-HS1). Numbers of detoxification genes are given in parentheses, and overlap classes are color coded. **c** For genes with *trans* eQTLs overlapping the HS1 window (genotype bins are in alternating gray and white, and bin midpoints in kb are indicated at bottom), log₂ fold changes (log₂FC) attributable to the respective genotypes at the *trans* eQTL loci at HS1 are shown on the y-axis. Within a bin, *trans* regulated genes (circles) are jittered to prevent plotting overlap, and are color coded after panel **b**. Detoxification genes are denoted with filled circles (legend, top left). **d** Scatter plot showing log₂FC values for differentially expressed genes (from **b**) in either the A-NIL-HS1[RR] versus A-NIL-HS1[SS] or the B-NIL-HS1[RR] versus B-NIL-HS1[SS] comparisons (x-axis and y-axis, respectively; genes are color coded based on **b**. **e** Relative expression levels for a subset of highly differentially expressed detoxification genes in NIL-HS1[RR] and NIL-HS1[SS] for the A and B NIL sets along with the respective values for F1s. To facilitate comparisons, expression was normalized to values from A-NIL-HS1[RR] and B-NIL-HS1[RR]. Significant differences are indicated by asterisks (adj-*p* < 0.05*, DESeq2; n.s., not significant). For the bar plot, means, error bars of ±2 standard error, and all data points are shown. The differential gene expression analyses for NILs in **b**, **e** used *p*-values adjusted for multiple tests (adj-*p* values) as output by DESeq2 (*n* = 5 biologically independent replicates, except for B-NIL-HS1[RR], for which *n* = 4 biologically independent replicates).

combinations were associated with low expression; Fig. 4b). This finding suggests that the action of the R strain HS1 *trans* factor(s) on *CYP392A12* is specific to the R haplotype at the *cis* eQTL (this potentially explains the finding with the HS1 NILs, which have the S genetic background except at HS1). To validate this conjecture, we constructed two sets of independent NILs in which the *CYP392A12* locus was introgressed from the R strain into the S strain for five generations (A-NIL-*CYP392A12*[RR] and B-NIL-*CYP392A12*[RR], along with the control lines A-NIL-*CYP392A12*[SS] and B-NIL-*CYP392A12*[SS]). We then crossed

these lines to the B-NILs we constructed for HS1, as shown in Fig. 4c, and performed reverse transcription quantitative PCR (RT-qPCR) for *CYP392A12* in resulting F1 females as well as those from the R and S parental strains. As expected, expression of *CYP392A12* was significantly higher in the R strain compared to the S strain (adj-*p* < 0.05, two-tailed t-tests with Bonferroni adjustment for multiple tests; Fig. 4d); further, intermediate expression of *CYP392A12* was otherwise only observed in F1 females when the RS genotype was present at both HS1 and *CYP392A12*, confirming the epistatic interaction.

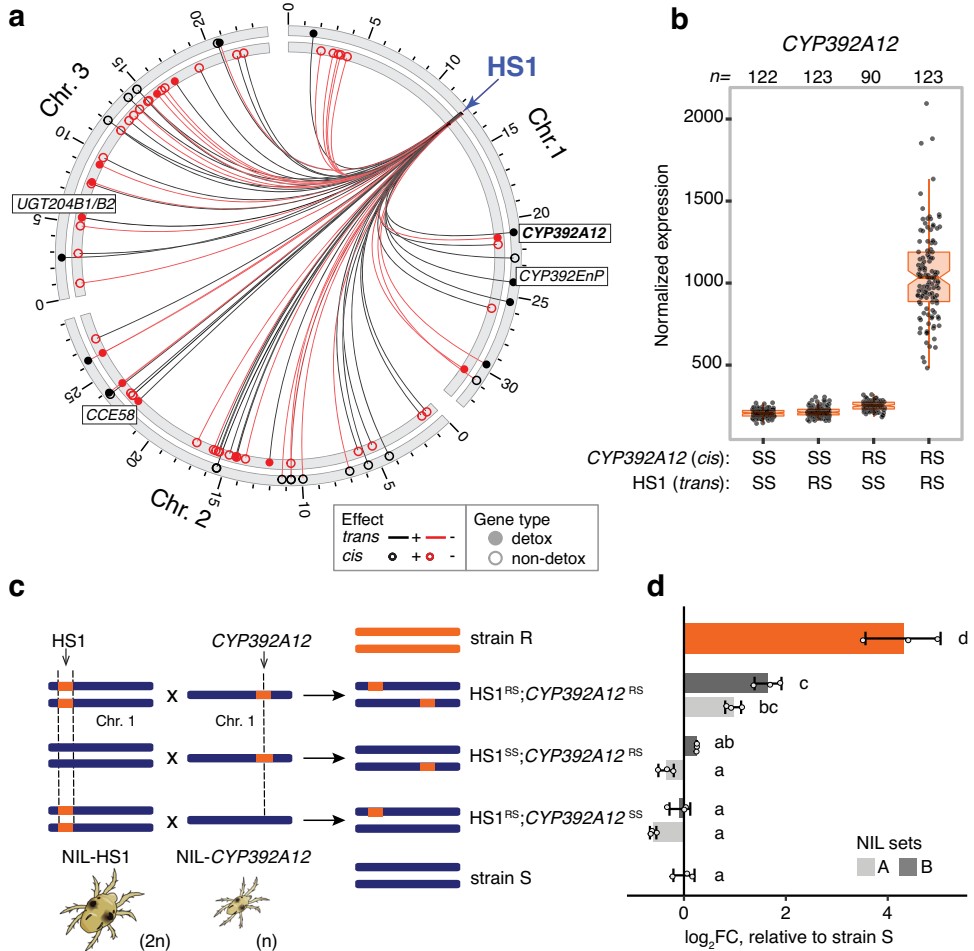

**Fig. 4 | An epistatic interaction between *CYP392A12 trans* and *cis* eQTLs. a** Genes *trans* regulated by HS1 that also have *cis* eQTLs are displayed in a circle plot (the lines originating from HS1, blue arrow, link HS1 to the genomic locations of the respective genes). Select detoxification genes (filled circles, legend, bottom right) with large HS1 *trans*-driven fold changes (*CYP392A12*, *UGT204B1*, *UGT204B2*, *CCE58*, and *CYP392EnP*, see Fig. 3c) are labeled, with *CYP392A12* in bold text. The direction of effects of *trans* eQTLs at HS1, as well as *cis* eQTLs at target genes, is as indicated in the legend; "+" indicates upregulation (black lines for *trans*, and outer circle for *cis*) by the RS genotype as compared to the SS genotype (otherwise, "-", red lines for *trans*, and inner circle for *cis*). **b** A boxplot showing DESeq2 normalized expression of *CYP392A12* across the 458 F3 families conditioned on the four possible genotypes at HS1 (reflecting a *trans* effect) and at *CYP392A12* (reflecting a *cis* effect; sample sizes, *n*, are given at top). For the boxplot, the box indentations are median values,

the boxes extend from the first to third quartiles, and the whiskers extend to 1.5 times the respective interquartile ranges. All data points are displayed. **c** Schematic showing crosses between the B-NIL-HS1 strain females (2n) and males (n) from the two sets ("A" and "B") of the *CYP392A12* NILs. RNA was collected from F1 females and from the parent R and S strains (color coded as orange and blue, respectively). **d** Displayed are log$_2$ fold change (log$_2$FC) values for *CYP392A12* in F1 females from the crosses shown in panel **c**, as well as strain R, in comparison to that of the S strain (one-way analysis of variance, $F_{7,16}$ = 82.67, $p$ = 2.29 × 10$^{-11}$; samples with different letters have significant differences as assessed with pairwise two-sided unpaired t-tests, adj-$p$ < 0.05; $p$ values were corrected for multiple tests with the Bonferroni method). For the bar plot, means, error bars of ±1 standard deviation, and all data points are shown ($n$ = 3 biologically independent replicates, with each biological replicate based on two technical replicates).

## Tandemly duplicated nuclear hormone receptor-96 (HR96) like genes at HS1

In the reference *T. urticae* genome sequence, 34 annotated genes are located in the HS1 interval (Supplementary Data 7). Previous functional genetic and molecular studies have identified genes in the aryl hydrocarbon receptor (AhR), cap' n' collar isoform C: Muscle Apo-neurosis Fibromatosis (CncC:Maf), and invertebrate HR96 (homologous to mammalian pregnane X, or PXR, and constitutive androstane receptors) families as regulators of responses to xenobiotics[19]. No *T. urticae* homologs of CncC:Maf or AhR are present in any of the HS1-9 intervals. However, one of the seven genes inclusive to the peak genotype bin for *trans* associations at HS1 (-12.507 Mb, Fig. 5a) is *tetur06g04270*. This gene encodes a product with homology to NHR proteins for which small molecule binding and (potentially) dimerization mediated by ligand-binding domains (LBDs) can lead to DNA-binding (mediated by DNA-binding domains, DBDs) to impact

transcriptional regulation[19,32]. Specifically, *tetur06g04270* is a homolog of the xenosensing NHR gene *HR96* in *D. melanogaster*[32]. As opposed to the single *HR96* gene in *D. melanogaster*, the reference *T. urticae* genome has eight genes encoding canonical HR96 products with both LBDs and DBDs, as well as 47 genes that encode products with HR96-like LBDs, but that lack DBDs[33]. One of the latter is *tetur06g04270*, hereafter called *HR96-LBD-1*. A *HR96* gene encoding both a LBD and DBD was present in each of the HS8 (*tetur20g01820*) and HS9 (*tetur11g01960*) intervals (Supplementary Data 7), but the genes regulated in *trans* by these hotspots are not strongly associated with detoxification (Supplementary Data 8).

As noted by Snoeck et al.[33], *HR96-LBD-1* is present as two copies in some *T. urticae* strains. Further, we observed that the coverage depth of R and S strain Illumina DNA reads aligned to *HR96-LBD-1* in the London reference genome was approximately twice that expected for single copy genes, a signature of a duplication (Supplementary Data 12

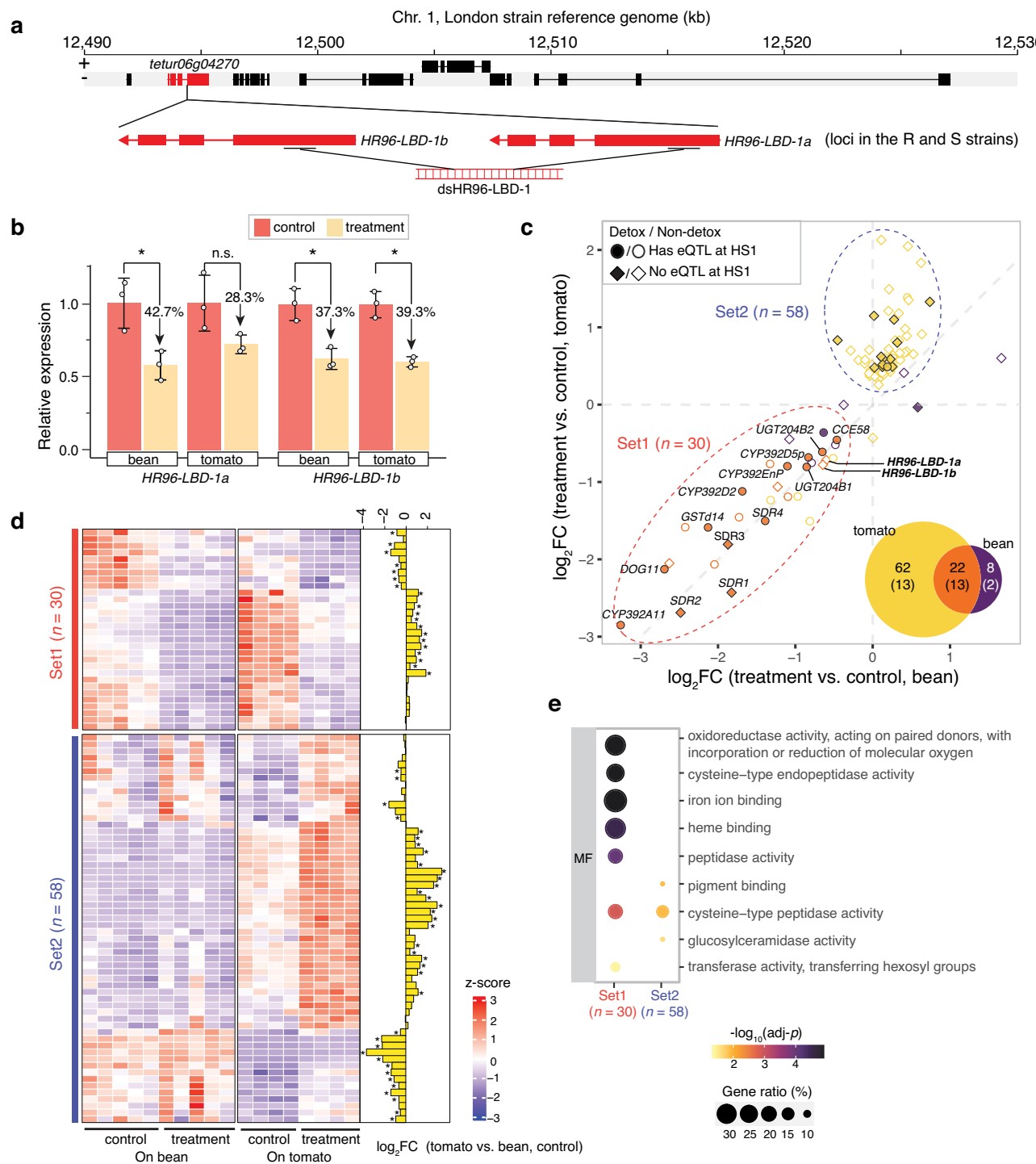

**a** Chr. 1, London strain reference genome (kb)

**b** control / treatment

**c** Detox / Non-detox; Has eQTL at HS1; No eQTL at HS1; Set2 (n = 58); Set1 (n = 30)

**d** Set1 (n = 30); Set2 (n = 58)

**e** MF; oxidoreductase activity, acting on paired donors, with incorporation or reduction of molecular oxygen; cysteine–type endopeptidase activity; iron ion binding; heme binding; peptidase activity; pigment binding; cysteine–type peptidase activity; glucosylceramidase activity; transferase activity, transferring hexosyl groups

and Methods). Therefore, we sequenced the R and S strain genomes using the long-read PacBio technology[34] and recovered and annotated the *HR96-LBD-1* locus from the resulting assemblies. In both strains, two genes, denoted *HR96-LBD-1a* and *HR96-LBD-1b* (Fig. 5a), are present as tandem duplicates (direct repeats with intergenic distances of 1.7 and 1.4 kb in the R and S strains, respectively, Supplementary Fig. 8 and Supplementary Data 13). The duplicates have 82.1% gap-compressed identity in coding regions in strain R at the amino acid level; the analogous value is 85.0% for strain S (Supplementary Fig. 9). While both genes harbor HR96-like LBDs, neither has a DBD. Substituting the *HR96-LBD-1* sequence in the reference genome for the PacBio assembled tandem duplicates allowed us to assess differential expression between strains for both *HR96-LBD-1a* and *HR96-LBD-1b*

with existing R and S strain RNA-seq data[18] (Supplementary Data 14; adj-$p$ < 0.01). While a small expression difference was observed between strains for *HR96-LBD-1a* (~60% expression in the R strain compared to the S strain), there was no significant difference for *HR96-LBD-1b*.

### RNAi knockdown of *HR96-LBD-1a* and *HR96-LBD-1b* alters detoxification gene expression

To assess if *HR96-LBD-1a* and *HR96-LBD-1b* impact the expression of genes *trans* regulated by HS1, we performed RNAi injections with dsHR96-LBD-1 to knock down both genes in B-NIL-HS1^RR (treatment; because the duplicated genes are highly similar, gene-specific RNAi knockdown sequences could not be designed, see Methods). We

**Fig. 5 | Effect of *HR96-LBD-1a* and *HR96-LBD-1b* knockdown on gene expression.** **a** Genes within the peak genotype bin interval for HS1 ("+" and "-", plus and minus strands, respectively; rectangles denote exons, and lines denote introns). The *tetur06g04270* (*HR96-LBD-1*; red) gene in the reference London strain genome (top) is present as two copies in the R and S strains (*HR96-LDB-1a* and *HR96-LDB-1b*, middle). A single dsRNA was selected that targets both duplicated genes (bottom; dsHR96-LBD-1). **b** Relative expression of *HR96-LBD-1a* and *HR96-LDB-1b* following injection with dsGFP (control) and dsHR96-LBD-1 (treatment) on the bean and tomato hosts as assessed with RT-qPCR (two-sided unpaired t-tests, adj-*p* < 0.05*; n.s., not significant; for tests from left to right: t(3.22) = 3.74, *p* = 0.04; t(2.44) = 2.45, *p* = 0.11; t(3.45) = 4.95, *p* = 0.02; and t(2.59) = 7.10, *p* = 0.02; *p* values corrected for multiple tests with the Benjamini-Hochberg method. For the bar plot, means, error bars of ±1 standard deviation, and all data points are shown (*n* = 3 biologically independent replicates, with each biological replicate based on two technical replicates). **c** Log₂ fold change (log₂FC) values for differentially expressed genes between treatment versus control samples for mites on bean (x-axis) and on tomato (y-axis), adj-*p* < 0.01 (multiple tests adjusted *p*-values are from DESeq2; *n* = 5 biologically independent replicates for mite samples on bean; *n* = 4 biologically independent replicates for mite samples on tomato). Each point is a gene color coded as shown in the Venn diagram (inset); detoxification genes are in parenthesis, and are displayed with filled plotting symbols, with shape denoting overlap to genes controlled in *trans* by HS1 (legend, top left). Select detoxification genes with large fold changes are labeled, as are *HR96-LDB-1a* and *HR96-LDB-1b*; genes with similar downregulation on both host plants are denoted as Set1 (*n* = 30); Set2 genes (*n* = 58) were upregulated upon treatment, mainly in mites feeding on tomato (dashed ellipses). **d** A heatmap shows standardized expression levels (z-scores) for genes belonging to Set1 and Set2 (each column is a biologically independent replicate, with the replicate numbers the same as for panel **c**). Log₂FC values for a pairwise comparison with control samples on tomato versus bean are displayed in a barplot at the far right (adj-*p* < 0.01*). **e** Enriched gene ontology (GO, based on molecular function, MF) term descriptions for Set1 and Set2 genes (adj-*p* < 0.05; *p*-values from one-tailed hypergeometric tests, followed by adjustments for multiple testing using the Benjamini-Hochberg method, as implemented in clusterProfiler). Circles are colored based on -log₁₀(adj-*p*) values, with size reflecting the number of observed genes with a term versus the number annotated with the term. The differential gene expression analyses in panels **c** and **d** (barplot) used *p*-values adjusted for multiple tests (adj-*p* values) as output by DESeq2.

performed two sets of RNAi injections using adult females, with injection of a sequence for green fluorescence protein as the control (dsGFP). For the first set of injections, mites were kept on bean leaves until RNA was harvested. To investigate a possible impact of host plant on detoxification and host associated gene regulation, for the second set injected mites were transferred from bean to tomato leaves for 24 h prior to RNA collection. Compared to bean, tomato is a challenging host for many *T. urticae* strains[26].

As assessed with RT-qPCR, with RNA-seq, or with both, we found that expression for each of *HR96-LBD-1a* and HR96-*LBD-1b* was significantly reduced by ~30–40% following dsHR96-LBD-1 injection as compared to the control (*p* < 0.05 for RT-qPCR, Fig. 5b; adj-*p* < 0.01 for RNA-seq, Fig. 5c, Supplementary Data 15). Further, as assessed with the RNA-seq data or by RT-qPCR (Supplementary Fig. 10), no other *HR96* genes in *T. urticae* were significantly changed in expression, suggesting that the knockdown was specific. In total, we identified 30 and 84 DEGs in response to the dsHR96-LBD-1 treatment for mites feeding on bean and tomato, respectively (Fig. 5c, Supplementary Data 11 and Supplementary Data 15). A subset of genes were downregulated in one or both comparisons (Set1, 30 genes; Supplementary Data 16) and were enriched for many of the same GO terms as observed for the genes *trans* regulated by HS1 (compare Supplementary Data 17 to Supplementary Data 8). These included most of the detoxification genes identified as strongly upregulated by the RS genotype at HS1 by eQTL mapping or in either the A-NIL-HS1^RR versus A-NIL-HS1^SS or B-NIL-HS1^RR versus B-NIL-HS1^SS comparisons (Supplementary Data 11); the most dramatically downregulated genes included *CYP392A11*, *DOG11*, *GSTd14*, and multiple SDRs (*SDR1-4* in Fig. 5c). Three of these SDRs (*SDR1-3*) are within ~500 kb of HS1, and by our eQTL classification criteria were assigned as having *cis* eQTLs (local associations; see Methods). However, the RNAi findings, along with their upregulation in the A-NIL-HS1^RR versus A-NIL-HS1^SS and B-NIL-HS1^RR versus B-NIL-HS1^SS comparisons (Fig. 3d, Supplementary Data 11), suggest that mechanistically *SDR1-3* are in fact *trans* regulated by the nearby *HR96-LBD-1* gene(s) located at HS1.

In contrast, a second group of genes (Set2, 58 genes; Supplementary Data 16) were primarily upregulated on tomato in response to knockdown treatment. Only a few of these genes were identified as under *trans* control by HS1 by the eQTL mapping that was performed on bean, and comparatively few were in annotated detoxification gene families (Fig. 3c; these genes were not enriched for detoxification-related GO terms, although they were enriched for a term associated with cysteine proteases, Fig. 5e and Supplementary Data 17). A differential expression analysis between the dsGFP injected mites feeding on tomato versus bean revealed that many Set1 and Set2 genes (63.3% and 63.8%, respectively) changed in expression upon host plant shift (adj-*p* < 0.01; *HR96-LBD-1b*, but not *HR96-LBD-1a*, showed minor upregulation, 1.3 fold, on the tomato host, Supplementary Data 15). Interestingly, a subset of Set2 genes that were induced upon transfer to tomato were upregulated even more strongly upon knockdown treatment, a contrasting pattern to that observed for most Set1 genes (Fig. 5d).

## Structural and allelic variation in *HR96-LBD-1a* and *HR96-LBD-1b*

To further examine variation in *HR96-LBD-1a* and *HR96-LBD-1b* in *T. urticae*, we aligned previously released Illumina genomic reads for 20 additional inbred *T. urticae* strains originating from Europe, Japan, and North America to the London reference genome sequence[18,35–37]. For all but one strain (C1N1d), normalized read coverage for the single *HR96-LBD-1* gene in the London genome sequence was similar to that observed for the R and S strains (Supplementary Data 12), suggesting the presence of two copies. For the C1N1d strain, zero coverage was observed, and a deletion of 9.6 kb that includes *HR96-LBD-1a* and *HR96-LBD-1b* was confirmed in a C1N1d genome assembly (see Discussion; Supplementary Fig. 8).

We also examined allelic variation in *HR96-LBD-1a* and HR*96-LBD-1b* beginning with the R and S strains for which we generated high-quality PacBio genome assemblies. At the amino acid level, the sequence identity for HR96-LBD-1a between the R and S strains is 91.5%, while it is 98.4% for HR96-LBD-1b. For HR96-LBD-1b, this included a radical tryptophan to arginine change at position 309 (W309R) in the R strain that was not observed in the S strain, or in either the R or S strains for HR96-LBD-1a (Supplementary Fig. 9). Alphafold-based modeling[38,39] and alignments of R strain HR96-LBD-1a and HR96-LBD-1b to other NHRs with known structures, including the human nuclear xenobiotic receptor PXR[40], revealed that although W309R is unlikely to be involved in the dimerization or activation potential of the LBD, it appears to be a key inward-facing residue in the bottom of ligand-binding pocket itself (Supplementary Fig. 11).

The high sequence similarity of *HR96-LBD-1a* and *HR96-LBD-1b* precluded confident construction of the complete sequences for these genes for the strains for which only short-read Illumina data were available. Nevertheless, the T-to-C transition that causes the W309R change (TGG to CGG) in the R strain was observed in Illumina read alignments for one other strain (the RB strain from Utah, USA), and a T-to-A transversion that also leads to the W309R change (TGG to AGG) was observed in four other strains (the MAR-ABi strain from Greece, and the Hib, KH and WG-S strains from Utah, USA; Supplementary Data 12). All variants predicted to cause the W309R substitution were inferred to be in *HR96-LBD-1b* as determined by nearby differences in Illumina reads fixed between *HR96-LBD-1a* and *HR96-LBD-1b* in the R and S strains, as well as duplicate-specific PCR and Sanger sequencing with the MAR-ABi strain (Supplementary Fig. 12).

## Discussion

Numerous studies have documented the upregulation of detoxification genes in insects and mites resistant to pesticides or responding to plant chemical challenges[6,22,41]. Although *cis* genetic variation has been a common explanation[42], *trans*-driven variation has also been reported[13–18]; however, critical questions about the genetic and molecular nature of *trans*-regulatory variation impacting detoxification and host plant use genes remain unanswered. In particular, what are the loci underlying *trans*-mediated variation in gene expression, and are they the same or different from those in metazoan xenobiotic response pathways established by molecular genetic studies?

We comprehensively identified loci explaining expression variation between a highly acaricide resistant *T. urticae* strain and a more susceptible one. Our finding of more *trans* than *cis* eQTL for all genes, as well as for detoxification genes, is explained in part by eQTL hotspots, such as HS5 on chromosome 2 (2.6–3.2 Mb, 1125 *trans*-regulated genes) that was also coincident with genotype ratio distortion in the eQTL mapping population, a signal of a segregating variant that impacts fitness (i.e., one that differentially impacts survival, as can be detected in multi-generational experimental designs). While a locus at HS5 may therefore impact fitness between the R and S strains, GO enrichment analyses suggested that this, and most other hotspots, did not contribute disproportionately to variation in detoxification gene expression. However, for the 182, 172 and 183 genes with *trans* associations to HS1 (chromosome 1, 12.4–12.6 Mb), HS6 (chromosome 2, 11.4–11.6 Mb) and HS7 (chromosome 3, 21.4–21.6 Mb), GO terms for detoxification were enriched, as they were for protease activity (HS1 and HS7), the upregulation of which is a potential mechanism to overcome anti-herbivore plant-produced protease inhibitors[30,43,44]. Among these three hotspots, HS1 was exceptional in both the percentage of detoxification genes with associations (30.8% versus 8.1% and 13.7% for HS6 and HS7, respectively, Supplementary Data 6), and in the magnitude of effect sizes explained in *trans*. In fact, as established with HS1 NILs, homozygosity of the R haplotype at HS1 was associated with changes of tens to > 200-fold elevated expression of subsets of detoxification genes in unrelated families. Where differences were observed between genes *trans*-regulated by HS1 and DEGs identified using HS1 NILs, potential explanations include epistatic effects, as observed for *CYP392A12*, highlighting the potential importance of interactions between *trans* and *cis* variants in the origin of expression variation in arthropod detoxification genes.

Among the transcriptional regulators commonly implicated in xenosensing and signaling in animals (i.e., CnC:Mafs, AhRs, and HR96 and its vertebrate homologs[19]), the number of *T. urticae* HR96 and HR96-like genes is striking. In particular, the large lineage-specific expansion of HR96-LBD genes[33] mirrors that of other families involved in responses to the environment, such as chemosensory receptors in animals or Resistance (R) genes in plants[45,46], and raises the possibility that HR96-LBD genes may be important for *T. urticae*'s ability to overcome the chemical defenses of its host plants. Supporting this conjecture, two HR96-LBD genes were located in the HS1 interval (*HR96-LBD-1a* and *HR96-LBD-1b*), and are causal for the major *trans* effects on gene expression mediated by HS1 as established by RNAi knockdown on two host plants. Further, genes regulated by one or potentially both of the two HR96-LBD genes included putative digestive proteases, as well as genes like *DOG11* (~ 63-fold upregulation by the RR haplotype), a member of a detoxification gene family with broad substrate specificity against plant-produced mono- and polycyclic catecholic compounds[25]. Further supporting a role for *HR96-LBD-1a* and *HR96-LBD-1b* in the regulation of host plant use associated genes, many of the genes that responded to RNAi treatment also changed expression upon host shift from bean to tomato.

*HR96-LBD-1a* and *HR96-LBD-1b* knockdown also impacted multiple CYPs in the CYP392 family, with the RR genotype in HS1 NILs resulting in the remarkably large upregulation of *CYP392A11* (~ 231-

fold), *CYP392D8* (~ 115-fold), and *CYP392D2* (~ 75-fold). Although roles for these CYPs in host plant interactions are not yet known, a combination of functional expression and genetic studies have revealed that CYP392A members metabolize structurally diverse acaricides such as pyflubumide, abamectin, cyenopyrafen and fenpyroximate[10,31,47]. Previously, with multi-generational evolve-and-resequence QTL mapping, Snoeck et al.[33] identified three resistance loci for the METI-Is acaricide tebufenpyrad in another inbred line derived from the MR-VP strain. One QTL localized to cytochrome P450 reductase, the required electron donor for microsomal CYP activities in animals, suggesting that CYP-mediated detoxification is important for metabolism of METI-Is compounds, and another localized to the *HR96-LBD-1a* and *HR96-LBD-1b* genic interval on chromosome 1. Collectively, these QTL findings[33] are consistent with a role for the R strain *HR96-LBD-1a* and *HR96-LBD-1b* haplotypes in resistance to tebufenpyrad, and potentially the other acaricides to which the R strain is highly resistant, via massive *trans*-driven upregulation of one or multiple CYP392 family genes.

No large differences in expression of *HR96-LBD-1a* or *HR96-LBD-1b* were observed between strains, suggesting that one or more coding sequence changes in *HR96-LBD-1a*, *HR96-LBD-1b*, or both, were involved in the upregulation of target genes in the R versus the S strain. For many NHR genes with both LBDs and DBDs, an interaction with an exogenous (xenobiotic) or endogenous ligand initiates translocation to the nucleus with homo- or heterodimer formation and DNA binding to alter transcription, although specific mechanisms vary[19,28]. Apart from HR96-LBD genes in *T. urticae*, several other instances of NHR genes with LBDs lacking DBDs have been reported for gene transcription regulation[28,48,49]. These potentially act by ligand- and LBD-dependent dimerization with canonical NHRs with both LBDs and DBDs to impact gene expression[48]. Within the LBDs of HR96-LBD-1a and HR96-LBD-1b, most changes were conservative, with the exception of one radical substitution (W309R) in the latter (Supplementary Figs. 9 and 11). Unexpectedly, we found independent mutational origins of the W309R change among geographically disparate *T. urticae* strains (Supplementary Fig. 12). This mirrors the finding of independent mutations with global occurrence for acaricide target-site resistance in *T. urticae*[50,51]. The six inbred strains with the W309R change all have known (or plausible) greenhouse origins (Supplementary Data 12), and therefore likely have recent histories of acaricide exposure. This includes MAR-ABi, which has an independent origin of the W309R change as compared to the R strain, and is also multi-acaricide resistant with high *trans*-driven expression of some of the identical genes regulated by HS1 in the R strain (e.g., *CYP392A12* and *DOG11*)[18].

Whether this single change, which is predicted to be inward facing into the putative ligand-binding pocket, is causal for the observed transcriptomic effects is still unclear. Regardless, this, or other changes within or outside the LBDs of HR96-LBD-1a or HR96-LBD-1b in the R strain, might enable the protein(s) to perceive an exogenous ligand (or a stress induced endogenous ligand) upon feeding on bean and tomato; an alternative possibility is that variant(s) in the R strain confer ligand-independent activation of target genes, which could explain the constitutive differences in detoxification gene expression between the R and S strains. While additional studies are required to assess these possibilities, as they are to establish which of the duplicate genes is causal, our work nevertheless establishes spider mite HR96-LBDs as master regulators of detoxification genes and other genes involved in host plant use. We also observed that a subset of genes that responded to RNAi knockdown of *HR96-LBD-1a* and *HR96-LBD-1b* differed between mites on bean versus tomato. This result suggests that signals from host plants, perhaps via plant specialized compounds, can modulate signaling by HR96-LBD proteins. We observed that 21 of the 22 *T. urticae* strains analyzed in our study appear to have both *HR96-LBD-1a* and *HR96-LBD-1b* (whether the

single copy in the London reference genome is an assembly error is currently unknown). Strikingly, however, both duplicates are absent in the C1N1d strain. This strain is unique among those we analyzed in that it originated from a well-characterized host-race specialist population of *T. urticae* that is restricted to European honeysuckle (*Lonicera periclimenum*) in the costal dune ecosystem in the Netherlands, where it exists in sympatry with generalist *T. urticae* populations[37]. Whether the absence of *HR96-LBD-1a* and *HR96-LBD-1b* is related to the specialist's restriction to honeysuckle is unclear. However, where generalist arthropod herbivores exhibit host-races or cryptic species complexes with host plant specialization (e.g., as for the whitefly *Bemisia tabaci*[52]), our work suggests that variation in xenobiotic regulators should not be overlooked as potential factors impacting host plant breadth.

In conclusion, in herbivores variation in regulatory pathways that putatively evolved in response to pressure by host plant factors (specialized compounds and proteins) is a likely target of selection by anthropogenic chemical application. Our findings raise the possibility that the lineage-specific expansion of HR96-LBD genes in *T. urticae* underlies modular control of subsets of xenobiotic response genes to enable productive interactions with the diverse host plants colonized by this cosmopolitan herbivore.

## Methods

### Mite strains and husbandry

The source and inbreeding of the multi-acaricide resistant strain MR-VPi (resistant strain, R) and the inbred strain ROS-ITi (susceptible strain, S), as well as their acaricide resistance profiles, have been described previously (Kurlovs et al.[18] and references therein). Briefly, the progenitor population (MR-VP) from which the R strain was inbred in 2018 was originally collected from a greenhouse in 2005 in Brussels, Belgium, and was maintained on bean plants (*Phaseolus vulgaris*) with periodic selection with a mitochondrial electron transport inhibitors of complex I (METI-Is) acaricide (tebufenpyrad). In contrast, the progenitor population from which the S strain was inbred in 2018 was collected on a rose species (genus *Rosa*) from a greenhouse in southern Italy in 2017 (see also Supplementary Data 12), and subsequently maintained on *P. vulgaris*. The R strain is moderately or highly resistant to bifenthrin (Na+ channel modulators, IRAC[53] class 3 A), fenbutatin oxide (inhibitor of mitochondrial ATP synthesis, IRAC class 12B), fenpyroximate, pyridaben, and tebufenpyrad (METI-Is, IRAC class 21 A), and cyenopyrafen (METI-IIs, IRAC class 20 A). The S strain is comparatively susceptible to all of these compounds, although it is moderately resistant to abamectin (GluCl allosteric modulators, IRAC class 6) and is resistant to dicofol, a compound of unknown mode of action. For propagation of bulk stocks and for collection of mites for DNA preparations, the strains were maintained on potted kidney bean plants (*P. vulgaris* var 'Prelude') at 25 °C (± 0.5 °C), 60% relative humidity, and a 16:8 h light:dark photoperiod in the absence of acaricide selection. Unless noted otherwise, for other experimental procedures mites were maintained under the same conditions on detached bean leaves.

### eQTL mapping population, RNA-seq generation, and read alignments

To map loci for expression variation between the R and S strains[54], we crossed S diploid virgin females (teleiochrysalis stage) to R haploid males. From F1 unfertilized daughters, we recovered recombinant F2 males and crossed them individually to ten S virgin females. We then collected 4-to-5-day-old adult F3 females from each cross (a median of 42 females per population). In total, 458 pools of F3 females (isogenic full sibling families) were collected and stored at -80 °C (a crossing and sample generation schematic is shown in Fig. 1a). RNA was extracted from the frozen F3 mite families using the RNeasy Plus Mini Kit (Qiagen, Germany) according to the manufacturer's Quick-Start Protocol. Quality and quantity of extracted RNA was analyzed by gel

electrophoresis (1% agarose gel; 30 min; 100 V) and a DeNovix DS-11 spectrophotometer (DeNovix, USA), respectively. Sequencing libraries were constructed using the Illumina Truseq stranded mRNA library preparation kit and sequenced on an Illumina Novaseq6000 to produce an average of 38.2 million paired-end reads of 100 bp per library (Supplementary Data 18). Library preparation and sequencing were conducted at Fasteris (Switzerland). RNA-seq reads from each F3 library were aligned to the three-chromosome London reference genome[36] using STAR v2.7.3a[55] with arguments of "--twopassMode basic --alignIntronMax 30000"; STAR was selected as it is mismatch tolerant, which reduces potential reference biases in read alignments to the *T. urticae* genome. Alignment BAM files were position sorted and indexed using SAMtools v1.9[56]. Unless otherwise noted, subsequent RNA collections and read alignments for downstream analyses used the same workflow as for the 458 F3 families.

### Variant predictions for the R and S strains

We aligned previously available Illumina DNA-seq reads for strain S (~ 442-fold genome coverage in paired-end reads of 151 bp) and strain R (~ 94-fold coverage in paired-end reads of 125 bp)[18] [PRJNA799176]) to the reference genome using BWA v0.7.17-r1188[57] with default options and predicted variants by adapting GATK v4.2 Best Practice recommendations[58]; hard filtering initial predictions with (1) RMSMappingQuality (MQ) ≥ 40.0, (2) StrandOddsRatio (SOR) ≤ 3, and (3) QualByDepth (QD) ≥ 2 identified 716,597 SNPs that distinguished the strains.

### Genotyping of isogenic F3 populations

For each of the 458 isogenic F3 populations at each SNP site (see Method section "Variant predictions for the R and S strains"), we counted the number of uniquely aligned RNA-seq reads originating from the R and S strains using a custom python script by employing Pysam v0.15.0[59] (https://github.com/pysam-developers/pysam). Based on the parental allele-specific RNA-seq read counts at SNP sites, we then assigned genotype calls (either heterozygous RS, or homozygous SS, the two possibilities given our experimental design; Fig. 1a). To do this, we retained only those sites with ≥ 5 reads supporting at least one parent; further, where non-parental bases were observed at SNP sites in reads (i.e., as can arise from sequence errors), we only retained sites for which reads supporting a non-parental base were < 5 or < 1% (between 17.5–23.2% of SNP sites were retained per F3 family). At these sites, the SS genotype was assigned when (1) the R allelic read count was < 5% of the sum of the R and S counts, and (2) the absolute number of R allelic counts was < 8. Otherwise, the RS genotype was assigned.

Subsequent to SNP-site level genotyping in each F3 population, which can be noisy (e.g., because of biases in allele-specific expression ratios resulting from *cis* variation), we assigned contiguous genomic intervals of RS or SS genotypes by assessing concordance of genotype calls among nearby SNPs (when multiple SNPs were present within a 100 bp interval, only one randomly selected site was used). Briefly, in tiling across chromosomes, when a change in the genotype at a SNP site was observed, if the new genotype was present at > 80% of genotyped sites in the downstream 250 kb interval, a change of genotypic state was introduced. The positions of recombination breakpoints were then assigned as the midpoints between the respective flanking junction SNP sites. Using the distance between the junction SNPs as a measurement for the resolution of recombination, ~81.0% (2372 of 2927) of recombination events were resolved to < 50 kb and ~50.0% to < 10 kb (Supplementary Data 1). To test whether recombination events were randomly distributed across individual chromosomes, we assessed the difference between the observed number of recombination events in sequential 1.5 Mb windows to the mean expectation per window based on the number of recombination events observed per chromosome. Significance of the deviation was assessed against a

distribution of the same metric assessed from 50,000 permutations with random recombination assignments by chromosome.

## eQTL mapping

For eQTL mapping, we selected 1889 marker loci distributed across the three *T. urticae* chromosomes in an approach adapted from Ranjan et al.[60]; briefly, marker loci positions were established as the midpoints between unique recombination events (genotype bins) inferred from the RNA-seq based genotyping of the 458 isogenic F3 families. Where no informative SNPs were present between nearby recombination events across multiple F3 families, the most 5' recombination event was used for bin construction (genotypes at each of the 1889 markers, Supplementary Data 2, for each F3 family were then imputed; a schematic illustrating genotype bin assignments is given in Supplementary Fig. 2). To generate expression phenotypes, we used htseq-count v2.0.1[61] on the RNA-seq alignments (parameters "-r pos -s reverse -nonunique none") for each of the 458 F3 families to count uniquely aligned reads per gene using the *T. urticae* GFF3 annotation reported by Wybouw et al.[36] to which we incorporated more recently manually curated genes available from the ORCAE database (v01252019 annotation)[62]. We then performed library size normalization of read counts using the estimateSizeFactors function in DESeq2 v1.34.0[63]; further, genes with raw read counts < 10 in more than 90% of F3 families were dropped from further analysis. Before eQTL association analysis, expression data were quantile normalized to ameliorate effects from outliers in gene expression. With the matched genotype and expression phenotype data, we performed eQTL mapping using the Matrix_eQTL_main function of MatrixEQTL v2.3[64] (parameters "pvOutputThreshold=0.01, pvOutputThreshold=0, useModel=model-LINEAR"). Associations with adj-*p* < 0.01 (false discovery rate in the MatrixEQTL output) were considered significant.

Because of linkage in the eQTL mapping population, a significant marker in a genomic interval will (typically) be flanked by adjacent markers that are also significant; additionally, genes can have multiple associations. To resolve and localize eQTL loci (i.e., identify markers with the most significant adj-*p* values while accounting for linkage), we first calculated the recombination fraction (rf) and logarithm of the odds (LOD) score between all pairs of marker loci using the functions est.rf and markerlrt in R/qtl v1.46[65]. Pairs of marker loci on the same chromosome for which rf <0.4 and LOD >3 were considered to be linked. For genes with significant association(s), the marker location with the lowest adj-*p* value by chromosome was taken as site of the association (the estimate of the causal locus location), and linked markers with less significant associations were removed from further consideration. This process was then iterated over the next most significant associations, if present, by chromosome. For retained association peaks, we required that the association be as significant, or more significant, than for 90% of the surrounding linked markers as determined with a rf < 0.3 and LOD > 20 (this final step was implemented to removed possible spurious associations arising from linkage that might not have been fully removed using the initial rf < 0.4 and LOD > 3 criteria).

## Classification of *cis* and *trans* eQTL and *trans*-QTL hotspot identification

From the distribution of distance of eQTL within 1.5 Mb from target genes, which are anticipated to be strongly enriched for *cis* eQTL, a background level was reached by about ±800 kb (Supplementary Fig. 3). Therefore, we classified associations within 800 kb of respective genes as *cis* eQTLs, and more distant ones as *trans* eQTLs. Hotspots for *trans*-eQTL were assessed using 200 kb non-overlapping windows across the genome; if a window originated 100 or more *trans* eQTLs, a hotspot was assigned (Fig. 2c, adjacent windows of > 100 eQTLs were merged). Genes regulated by hotspots were recovered from genotype bins overlapping the respective window(s).

## Detoxification genes and their potential transcriptional regulators

A set of *T. urticae* genes belonging to families associated with the metabolism, binding, or transport of xenobiotics (detoxification genes, Supplementary Data 5) were adapted from Kurlovs et al.[18]. Genes in *T. urticae* encoding products with homology to HR96 and CncC:Maf proteins were previously annotated by Snoeck et al[33]. (55 genes) and Dermauw et al[22]. (*tetur07g06850* and *tetur07g04600*), respectively. To identify homolog(s) of AhR in *T. urticae*, we performed a Blastp search (E-value < 1e-10) against the *T. urticae* proteome with the *D. melanogaster* AhR protein Spineless (Ss; FBpp0297169, Flybase[66]), and resulting *T. urticae* hits were used in reciprocal Blastp searches of the *D. melanogaster* proteome. A single protein with a reciprocal best hit to Spineless was retained as a putative *T. urticae* AhR ortholog (product of *tetur03g01600*). Protein identifiers and sequences were recovered, and Blastp searches performed, with Flybase (version FB2023_02) and ORCAE[62] accessed on 24 May, 2023.

## Construction and characterization of NILs for HS1

With seven rounds of recurrent backcrossing we introgressed the R haplotype at the HS1 region on chromosome 1 at ~12.5 Mb (see Results) into the S genetic background to generate two independent NILs, A-NIL-HS1^RR and B-NIL-HS1^RR. Each line originated from a different F0 cross of a virgin S female to an R male, with recurrent backcrossing to S females. During backcrossing, Cleaved Amplified Polymorphic Sequences (CAPS) markers developed using R and S variant predictions at and nearby the HS1 interval were used to select for the R haplotype, and to identify recombination events immediately flanking the HS1 interval. For the final recurrent cross, the S haplotype at HS1 was also selected to produce two control lines for each NIL, denoted A-NIL-HS1^SS and B-NIL-HS1^SS. For the marker assisted backcrossing, DNA from single mites were extracted[67] and used as template for PCR reactions using the GoTaq® DNA Polymerase kit (Promega, USA) following the manufacturer's instructions in a total reaction volume of 25 µl with CAPS marker specific primers. For restriction digests, PCR reactions were supplemented with 20 units of XbaI, HaeIII or MspI (NEB, USA) in a volume of 25 µl of 1× Cutsmart buffer, incubated overnight at 37 °C, and products were resolved on 2% agarose gels. For each CAPS marker, the respective location, primers, restriction enzyme, and expected banding pattern by genotype is given in Supplementary Data 19.

To resolve the boundaries of the R haplotype present at HS1 in A-NIL-HS1^RR and B-NIL-HS1^RR, as well as to assess differential gene expression between these NILs and their matching control lines (A-NIL-HS1^SS and B-NIL-HS1^SS, respectively), we generated RNA-seq data for each line with 5-fold biological replication. Additionally, we generated matching RNA-seq data for respective F1s derived from crosses of A-NIL-HS1^RR females to A-NIL-HS1^SS males, and B-NIL-HS1^RR females to B-NIL-HS1^SS males. On average, 46 4-to-5-day-old female mites were used for each genotype and biological replicate. RNA collection and RNA-seq read alignment and gene expression quantification was done as for the 458 F3 families used for eQTL mapping. For pairwise comparisons, differential gene expression was detected with DESeq2 v1.34.0[63] (adj-*p* < 0.01, absolute log2FC > 0.5, and lfcSE < 1). Only biological replicates with coefficient values of $R^2$ > 0.9 were included in pairwise comparisons (one B-NIL-HS1^RR replicate was removed). Finally, for each replicate for each NIL RNA-seq genotyping as described in Methods section "Genotyping of isogenic F3 populations" was adapted to refine the breakpoints of the R strain haplotype at HS1 in A-NIL-HS1^RR and B-NIL-HS1^RR, as well as to assess residual R sequences genome-wide (Supplementary Fig. 7).

## Construction and characterization of NILs for *CYP392A12*

Using the same experimental procedure as for the construction of NILs at HS1, we generated independent NILs in which the *CYP392A12* locus

from the R strain was introgressed with five backcrosses into the S genetic background. The resulting NILs, A-NIL-CYP392A12[RR] and B-NIL-CYP392A12[RR], and their matching control lines, A-NIL-CYP392A12[SS] and B-NIL-CYP392A12[SS], were confirmed to have the respective genotypes at *CYP392A12* and to have the S strain genotype at HS1 (CAPS markers used for introgression and genotyping at HS1 are given in Supplementary Data 19). To understand the impact of the R genotype at HS1 on *CYP392A12* expression, we crossed males for the NILs for *CYP392A12* to B-NIL-HS1[RR] and B-NIL-HS1[SS] females (Fig. 4c). With three biological replicates per cross, we collected 100-120 resulting 4-to-5-day-old F1 female mites, extracted RNA, and determined expression of *CYP392A12* by RT-qPCR. Synthesis of cDNA was performed using the Maxima First Strand cDNA Synthesis Kit (Thermo Fischer Scientific, USA) with RT-qPCR reactions conducted using the GoTaq® qPCR Master Mix (Promega, USA) with the primers listed in Supplementary Data 20 in a Mx3005P qPCR machine (Agilent Technologies, Belgium). Cycle conditions were 95 °C for 10 min followed by 40 cycles of 95 °C for 15 s, 55 °C for 30 s, and 60 °C for 30 s, followed by a melting curve analysis step. Melting curves and no-template controls (NTC) were used to confirm, respectively, the specificity of amplification and absence of contamination. A serial dilution of pooled cDNA was used to determine the mean amplification efficiency of each gene-specific primer pair (values of 1.9 to 2 were considered acceptable). The program qbase+ (Biogazelle, Belgium) was used for the analysis of raw quantification cycle (Cq) values, which were all first normalized against the housekeeping genes *ribosomal protein 49* (*Rp49*, tetur18g03590) and *ubiquitin C* (*UBQ*, tetur03g06910). Two technical replicates were used for each biological replicate, and mean $\log_2$FC values relative to the S strain were assessed along with respective standard deviations. Statistical analyses were performed using one-way analysis of variance followed by two-tailed unpaired t-tests with Bonferroni correction for multiple testing.

### Annotation and expression of *HR96-LBD-1a* and *HR96-LBD-1b*
To characterize the *HR96-LBD-1* (*tetur06g04270*) locus at HS1 (see Results), high-molecular-weight DNA was extracted from both the R and S strains with a protocol adapted from the Qiagen genomic 20/G Tip kit (Qiagen, Germany). Specifically, viable mites transferred in bulk to the leaves of twelve bean plants were collected using a mite brushing machine (Leedom Enterprise, USA) directly into a prechilled, ice-cold mortar. The mites were ground in liquid nitrogen and subsequently divided into two prechilled microcentrifuge tubes. Next, 1.5 ml lysis buffer (20 mM EDTA, 100 mM NaCl, 500 mM guanidine-HCl, 10 mM Tris, 1% Triton-X), 6 µl RNase A (20-40 mg/ml) and 30 µl proteinase K (10 mg/ml) were added to each tube. Following an incubation of 30 min at 37 °C with gentle agitation, another 60 µl proteinase K and 3 µl RNaseA were added, and each tube was incubated for an additional 2 h at 50 °C with gentle agitation. The samples were then centrifuged (20 min, room temp, 13000 rcf) to pellet the debris. After equilibration of the 20/G Tip with 1 ml QBT buffer, the clarified lysate was pooled again and transferred to a 20/G Tip and allowed to drain by gravity; the 20/G tip was washed 4 times with 1 ml QC buffer and subsequently eluted with 1.2 ml pre-warmed (50 °C) QF buffer (Genomic DNA Buffer Set, Qiagen). Genomic DNA was then precipitated, ethanol washed and resuspended in nuclease free water[68]. From each sample, sequencing libraries were constructed with the Pacbio SMRTbell NGS Library Preparation kit and sequenced on a PacBio Sequel instrument (VIB Nucleomics Core, Leuven, Belgium).

Resulting PacBio reads were assembled using the default settings of Flye v2.5[69] with the target genome size set to 90 Mb. Subsequently, Illumina genomic reads available for each strain[18] were aligned to the resulting assemblies using the default settings of BWA-MEM v0.7.17-r1188[57] and sorted by position using SAMtools v1.9[56]. The default settings of Pilon v1.22[70] were then used to polish each assembly. From the resulting assemblies (scaffold N50 values of 14.1 and 5.4 Mb for the R

and S strains, respectively), we recovered the *HR96-LBD-1* region from both the R and S strains with Blastn and tBlastn v2.6.0 + [71] searches with reference genome *HR96-LBD-1* sequences as query. Two copies of the respective gene in each strain, denoted *HR96-LBD-1a* and *HR96-LBD-1b*, were then manually annotated using GenomeView vN42[72]. To assess expression of each gene copy, the *HR96-LBD-1* locus (±1 kb) in the *T. urticae* reference genome was masked, and the respective tandemly duplicated *HR96-LBD-1a* and *HR96-LBD-1b* intervals for each of the R and S strains were appended to the genome sequence. RNA-seq reads from the R and S strains[18] (PRJNA801103) and RNAi knockdown samples were then aligned to the modified genomes for differential gene expression analyses (DESeq2 v1.34.0[63], adj-$p$ < 0.01).

### *HR96-LBD-1a* and *HR96-LBD-1b* RNAi knockdown and detection of differentially expressed genes
Primer pairs designed with Primer3[73] that incorporated a T7 promoter were used to amplify (1) both *HR96-LBD-1a* and *HR96-LBD-1b* sequences (297 bp for each) from cDNA of strain B-NIL-HS1[RR] and (2) a GFP sequence from plasmid DNA (454 bp; also see Supplementary Data 21). For primer selection for *HR96-LBD-1a* and *HR96-LBD-1b*, si-Fi v21_1.2.3-0008[74] was used to minimize potential off-target effects; it was not possible to design dsRNA probes specific to each of the duplicated *HR96-LBD-1a* and *HR96-LBD-1b* genes (i.e., the same primer pair amplified both). PCR was performed with the Expand™ Long Range dNTPack (Roche, USA) reagents following manufacturer's instructions with 2 min at 92 °C, five touch-down cycles of denaturation at 92 °C for 20 s, annealing at 60 °C -1 °C/cycle for 20 s and elongation at 68 °C for 1 min, followed by 37 cycles of 92 °C for 20 s, 55 °C for 20 s and 68 °C for 1 min, and finally 68 °C for 5 min. Next, dsRNA was produced with the TranscriptAid T7 High Yield Transcription Kit (Thermo Fisher Scientific, USA) according to the manufacturer's instructions with 1 µg of T7 products as templates in 20 µl reactions incubated overnight at 37 °C. Template DNA was then degraded by DNase treatment (TURBO DNA-free™ Kit, Invitrogen, USA), dsRNA was recovered by chloroform-phenol extraction, and concentrations were evaluated using a DeNovix® DS-11 FX spectrophotometer (DeNovix, USA).

With dsRNAs diluted to 1 µg/µl with nuclease-free water (Integrated DNA Technologies, USA), 150 2-to-3 day old adult female mites of the B-NIL-HS1[RR] strain were injected as previously described[75] under a Leica S8 APO microscope (Leica Microsystems, Germany) with a Nanoject III microinjector (Drummond Scientific, USA) with needles pulled from 3-000-203-G/X Glass Capillaries (Drummond Scientific, USA) with a P-1000 Micropipette Puller (Sutter Instruments, USA) with settings "heat: 500, pull: 60, velocity: 70, delay: 200, pressure: 500, Ramp: 490". Needles were sharpened with a BV-10 Micropipette Beveler (Sutter Instruments; 15° angles). Each female was injected with 3 nl near the third pair of legs. After injecting, mites were placed on detached bean leaves on wet cotton and allowed to recover.

For RNA extraction of mites on bean, 50–100 mites that survived injections with dsRNA against *HR96-LBD-1a* and *HR96-LBD-1b* (treatment) and *GFP* (control) were collected at four days with five biological replicates each. For a separate set of injections, all surviving mites were maintained on bean for three days before being transferred to tomato (*Solanum lycopersum* var. 'Moneymaker') for 24 hours prior to collection (four biological replicates). From resulting RNA-seq alignments DEGs were detected with DESeq2 v1.34.0[63] (adj-$p$ < 0.01). Using three of the biological replicates each for the mites maintained on bean or transferred to tomato, the efficiency of the RNAi knockdown of *HR96-LBD-1a* and *HR96-LBD-1b* was assessed by RT-qPCR with primer pairs designed to be specific for each gene duplicate. Additionally, the expression of *tetur86g00030*, the most closely related gene to *HR96-LBD-1a* and *HR96-LBD-1b* as predicted by si-Fi v21_1.2.3-0008[74], was assessed by RT-qPCR to evaluate possible RNAi off-target effects. Methods used for RT-qPCR were the same as used for *CYP392A12*; primer sequences are in Supplementary Data 20. Statistical analyses of

RT-qPCR data were performed using two-tailed unpaired t-tests followed by the Benjamini-Hochberg method to adjust for multiple tests.

## GO enrichment analyses for specified gene sets

GO enrichment analyses were performed with the "enricher" function of clusterProfiler v4.2.2[76] (parameters "pAdjustMethod = 'BH', pvalueCutoff = 0.05") using Molecular Function (MF) terms available from the ORCAE database (v01252019)[62]. For analyses of genes in HS1-HS9 (Fig. 2c), those with raw read counts < 10 in more than 90% of the 458 eQTL F3 families were not included in the universal background gene set; for the analyses of genes identified by RNAi knockdown of *HR96-LBD-1a* and *HR96-LBD-1b*, the universal background set consisted of genes with read counts > 10 in all respective samples.

## HR96-LBD-1a and HR96-LBD-1b alignments, homology modeling of HR96-LBD-1b, and alignment with known LBD structures

Multiple sequence alignments with HR96-LBD-1a and HR96-LBD-1b proteins were constructed using MAFFT v7.505[77] with "--clustalout". To predict domains, we used InterProScan v91.0[78] with InterPro protein signature databases[79]. Additionally, ColabFold v1.5.0[39] was used for homology modeling of HR96-LBD-1b. This program combines the fast homology search capacity of MMseqs2[80] with the highly accurate protein structure prediction of AlphaFold2[38]. Amber relaxation was applied to remove distracting stereochemical violations of the model without the loss of accuracy. Subsequently a partial alignment covering the region ranging from AA 251–444 of HR96-LBD-1b that could be predicted with high confidence was made using PROMALS3D[81] (http://prodata.swmed.edu/promals3d, accessed on 1 May, 2023, running PROMALS3D v1), including also HR96-LBD-1a, HR96 of *Drosophila melanogaster* (NP_524493.1) and the protein sequences of other NHRs with known crystal structures: RXRalpha (pdb_6hn6[82]), CAR (pdb_1XV9[83]), PXR (pdb_6XP9[84]) and VDR (pdb_1DB1[85]) of human origin and daf12 of *Strongyloides stercoralis* (pdb_3GYU[86]).

## Characterization of *HR96-LBD-1* copy number in a global collection of *T. urticae* strains

We characterized copy number variation of *HR96-LBD-1* in 22 inbred *T. urticae* strains (including the R and S strains) where Illumina genomic read data were available[18,35–37] (PRJNA387043, PRJNA498683, PRJNA530192, PRJNA597924, PRJNA799176); a summary of strain information is provided in Supplementary Data 12. Previously, we used read cover per gene, normalized to the genome-wide coverage depth, to identify copy number variation for several *T. urticae* genes[10,36]. To estimate the copy number of *HR96-LBD-1*, we adapted these methods by first aligning reads from the 22 strains to the London reference genome with BWA v0.7.17-r1188[57] followed by sorting and indexing with SAMtools v1.9[56]. For each strain alignment BAM file, the median per base coverage of coding bases in *HR96-LBD-1* was then divided by the median of coverage depth at bases in the other coding genes in the *T. urticae* genome (with this analysis, the normalized coverage depth for a single copy gene is expected to be ~1, for a duplicated gene ~2, etc.). These analyses were performed with a custom script that used Pysam v0.15.0[59], and only primary alignments were used ("flag_filter=256, min_mapping_quality=0").

Because zero read coverage at *HR96-LBD-1* was observed for strain C1N1d (Supplementary Data 12), we generated a de novo assembly of this strain (scaffold N50 of 48.3 kb) using the existing Illumina read data[37] and SOAPdenovo2 v2.4[87] (command "SOAPdenovo-63mer" with parameters "all -K 33 -p 50" and configuration file flags "max_rd_len=125 avg_ins=550 reverse_seq=0 asm_flags=3 rd_len_cutoff=120 rank=1 pair_num_cutoff=3 map_len=32"). With Blastp v2.9.0 +[71] searches with genes located upstream (*tetur06g04250* and *tetur06g04260*) and downstream (*tetur06g04290* and *tetur06g04300*) to the *HR96-LBD-1* locus in the London genome sequence, we recovered a scaffold spanning the respective genomic interval in the C1N1d strain and

aligned it to the syntenic regions from the R and S strain genomes with MAFFT v7.505[77] (genes internal to a deletion including *HR96-LBD-1a* and *HR96-LBD-1b* in the C1N1d strain in the aligned sequences were manually annotated, Supplementary Fig. 8).

## Allelic variation in HR96-LBD-1a and HR96-LBD-1b underlying the W309R change

We used the Illumina DNA-seq alignment data for the 21 inbred strains (except C1N1d) and Pysam v0.15.0[59] to recover reads spanning position 12,494,172 on chromosome 1 in the London reference genome that leads to the W309R change in the R strain HR96-LBD-1b (see Results; normalized per base coverage depths were also calculated as for the gene-level copy number analysis, see Supplementary Data 12). For each strain alignment file, we then stratified the reads by nucleotide when base variation was observed at position 12,494,172, and for each of the two resulting read sets we generated de novo assemblies with SOAPdenovo2 v2.4[87] to recover the ~90 bp up- and downstream of the position (command "SOAPdenovo-63mer" with parameters "all -K 25 or 27 -R" and configuration file flags "max_rd_len=125 asm_flags=3 rd_len_cutoff=75 map_len=15"). In addition to the T or T and C nucleotides at position 12,494,172 observed in S and R strain aligned reads, respectively, we observed T and A nucleotides at this position in several strains including the MAR-ABi strain. To validate the A variant, we used the R and S strain PacBio-assembled genomes to design *HR96-LBD-1a* and *HR96-LBD-1b* specific primers (Supplementary Data 22) and amplified and Sanger sequenced the duplicated genes using R and S strain DNA (as a control) and DNA obtained from a MAR-ABi strain female extracted as described by Bajda et al.[67]. PCR reactions were performed using the GoTaq® DNA Polymerase kit (Promega, USA) following the manufacturer's instructions in a total reaction volume of 30 μl, and Sanger sequencing was performed at LGC Genomics (Teddington, UK) using the original PCR primers.

The *HR96-LBD-1a* and *HR96-LBD-1b* sequences of the R and S strains (PacBio assemblies), along with the sequences obtained by de novo assemblies and PCR and Sanger sequencing, were aligned using MAFFT v7.505[77]; six fixed nucleotide differences between *HR96-LBD-1a* and *HR96-LBD-1b* flanking the DNA codon for position 309 were used to assign sequences to each duplicate (Supplementary Fig. 12).

## Statistical analyses and display items

Unless noted otherwise, statistical analyses were performed in R v4.1[88]. A heatmap was generated using ComplexHeatmap v2.10[89] and Venn diagrams using VennDiagram v1.7[90]. Other plots were made using ggplot2 v3.3[91], and adjusted as needed in Adobe Illustrator (Adobe, CA, USA).

## Reporting summary

Further information on research design is available in the Nature Portfolio Reporting Summary linked to this article.

## Data availability

Data supporting the findings of this work, including eQTL mapping and differential gene expression analyses, are available within the paper and its Supplementary Information files. RNA-seq reads and gene expression metadata have been deposited at Gene Expression Omnibus (Project GSE221677). The S (genome version JAPRAR000000000) and R (genome version JAPRAS000000000) assemblies, along with the respective PacBio DNA reads, have been deposited to National Center for Biotechnology Information, NCBI (BioProjects PRJNA907360 and PRJNA907031, respectively). The C1N1d strain genome (genome version JASKHX000000000) assembly has been deposited to NCBI under the previously published BioProject PRJNA597924. Sanger sequences and targeted Illumina assemblies of *HR96-LBD-1a* and *HR96-LBD-1b* have been deposited at GenBank (accessions OR067932 to OR067949) and genetic marker data used

for eQTL mapping are provided on FigShare[92]. The assembly of the C1N1d strain used previously published Illumina DNA read data (NCBI BioProject PRJNA597924), and previously published Illumina DNA read data were also used for *HR96-LBD-1a* and *HR96-LBD-1b* copy number analyses (NCBI BioProjects PRJNA387043, PRJNA498683, PRJNA530192, PRJNA597924, and PRJNA799176) and for targeted de novo assemblies (NCBI BioProjects PRJNA530192 and PRJNA799176). RNA-seq data used for expression studies with the S and R strains were published previously (NCBI PRJNA801103). Previously published protein sequences, or structures, that supported HR96-LBD-1 alignments included NP_524493.1 (NCBI), pdb_6hn6 (Protein Data Bank, PDB), pdb_1XV9 (PDB), pdb_6XP9 (PDB), pdb_1DB1 (PDB), and pdb_3GYU (PDB). Source data are provided with this paper.

## Code availability
Custom scripts used in the analysis are available on Github (https://github.com/rmclarklab/mite_eQTL; https://doi.org/10.5281/zenodo.7992545).

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

## Acknowledgements
We thank Dr. Robert Greenhalgh for assistance with genomic analyses. This research has received funding from the European Research Council (ERC) under the European Union's Horizon 2020 research and innovation programme (Grant agreement No. 772026-POLYADAPT and 773902-SuperPests), the Special Research Fund of Ghent University (grant BOFSTA2017003701) and the Research Foundation Flanders (grant G035420N) to T.V.L.

## Author contributions
R.M.C., T.V.L. conceived and designed the study; R.M.C. and T.V.L. supervised the experiments; M.V., B.D.B., S.D.R., E.V.P. conducted the experiments; M.J., M.V., R.M.C., T.V.L., R.F. analyzed the data; M.J., R.M.C., B.D.B., M.V., R.F., T.V.L. drafted the original manuscript; we critically revised the manuscript and approved the final version.

## Competing interests
The authors declare no competing interests.
