## [Peer Review File · Nature Communications]

Reviewers' Comments:

Reviewer #1:

Remarks to the Author:

This is a very interesting study that leverages some of the unique aspects of the *Tetranychus urticae* system to obtain significant new results on trans-acting factors regulating families of detoxicative genes in this agricultural pest. The work started with two different inbred strains, one with high constitutive expression of many putative detoxicative genes. A crossing design produced haploid F2 males carrying recombinant chromosomes, which were individually crossed to females of the other strain, producing identical F3 daughter cohorts to generate enough tissue for RNA-seq analysis. Then the males (or their daughter cohorts) were genotyped to identify recombinant chromosomal regions that influenced differentially-expressed genes identified by RNA-seq. cis- or trans-acting factors could thus be identified by comparing the chromosomal region with the location of the differentially expressed gene. One especially influential trans-acting region was investigated in more detail, and found to contain a tandem duplication of an HR96 gene. RNA interference of both duplicated genes had effects on target detoxicative genes consistent with the eQTL analysis.

The system is very impressive and the results are quite rich and complicated. Although the results do not reveal a novel family of transcriptional regulators, they are important in establishing the universality of the HR96 genes. However, from a mechanistic point of view, the results are frustratingly incomplete. Since the identified HR96 genes lack DNA-binding domains, indirect effects due to dimerization with other unknown family members with DNA binding domains must be hypothesized. Since it was not possible to target each duplicate HR96 gene separately with RNAi, the significance of the "radical tryptophan to arginine" substitution is not resolved. Perhaps the authors could check to see whether this residue occupies a site in the dimerization domain-- this would not require additional experiments. This ambiguity also hampers understanding of the very interesting epistatic interaction between the trans-acting transcription factor and the cis-acting regulatory sequence. Perhaps the authors could probe the difference between the R and S strains in the cis-factor--this would not require additional experiments. A more complete mechanistic understanding would require different approaches; the authors have pushed the genetic approach as far as it can go. This should not stand in the way of publication, as the study nicely illustrates the limitations as well as the strengths of the genetic approach in the fascinating and economically important *Tetranychus* system.

The manuscript should be rewritten for a general audience. The style is dense and highly compressed, requiring that a sentence be read several times to unpack the meaning. The frequent use of abbreviations and acronyms is convenient for the authors but not for the readers. Each paragraph in the Results would benefit from an introductory sentence motivating the results, instead of leaving it to the reader to deduce the motivations after having read the entire paragraph.

Reviewer #2:

Remarks to the Author:

Review of Ji et al. on the identification of a gene underlying large trans-effect upregulation of detoxification and host plant use genes in a generalist herbivore.

The authors created 458 F3 backcross females, which were subsequently used in eQTL mapping, by generating three generation crosses of *Tetranychus urticae*, by firstly mating an acaricides resistant with a susceptible strain; the resulting F1 females were kept unmated to produce hybrid recombinant sons (F2); then, these haploid F2 males were backcrossed into females from the susceptible strain, which ultimately generated the diploid F3 backcross females. Information on genotype data (using 1889 informative genotype bins) with information on gene expression phenotypes were combined to identify (using a linear model) 5,685 local associations (cis-eQTLs, with 54.7% within +/- 100kb) and 10,563 gene expression associations in trans (trans-eQTLs). A total of 9,740 genes were found associated with eQTLs when exclusively considering those genes located on the chromosome assembly of the reference genome of *T. urticae*. A considerable number (537) of genes with eQTLs were identified that are of interest for specialized metabolism,

transport and detoxification in explaining resistance and host use. Several eQTL hotspots with an accumulation of trans-eQTLs were identified, of which, a hotspot termed HS1 was further investigated. To do this, near-isogenic lines were created. The resistance haplotype at hotspot HS1 was introgressed into the susceptible genetic background. Then, the authors searched for an overlap between differentially expressed genes between those lines and the trans-regulated genes at HS1 from the eQTL study. Resistance was also found related to higher expression at the CYP392A12 locus when the resistance allele was present (this finding was validated by introgressing this locus from the resistant into the susceptible *T. urticae* strain to create an RS genotype and comparing expression levels for this locus with the parental strains, SS or RR at that locus). Further investigation of the HS1 was done by exploring the genomic region underlying this hotspot. To do this, long-read sequencing was used to improve this genomic region's assembly, where HR96-like genes were identified to be in fact tandemly repeated. The influence of those two tandemly repeated genes on the trans-regulation of genes by HS1 was checked by jointly knocking both loci down in an RNAi experiment, and studying resultant transgenic *T. urticae* lines for their performance on bean and tomato hosts, with tomato the more challenging host for this generalist herbivore. Many detoxification genes that were otherwise upregulated in resistant (RR-HS1) versus susceptible (SS-HS1) near-isogenic lines were downregulated in RNAi background of HR96-like genes (those previously found within HS1 location from the eQTL analysis).

In sum, the authors presented a very interesting work related to an important topic that is the biology and evolution of herbivory. They used genetical genomics to infer important regions on the genome of a generalist, regions that are potentially related to the herbivore's host plant use potential. Several lines of evidence suggested that important genetic determinants of host plant use metabolism were unravelled. However, two points could be added to the analyses: 1) integrate phenotypic data that define acaricide resistance and identify collocation with eQTLs, and thus, improve information about genes that directly underlie this specific resistance trait; 2) an analysis of the hierarchical order of gene regulation based on trans-eQTLs for each gene to deduce a 'global regulatory hierarchy' in resistance. Also, I found the Discussion limited (with sometimes redundant information provided); it could be improved by providing more in-depth insights for certain points (comments provided below).

Other comments:

L25: please specify these additional approaches (reverse genetics?)

L56-57: what is the chemical basis of these pesticides?

L65: name a few of such genes here

L74: where do these lines originate from?

L79-80: more specific, numbers?

L79: remove 'the' before 'most'.

L80-81: are you talking here about copy number variation since they are tandemly duplicated, see L78/9?

L83: 'intra-specific variation in regulators': this should be more specific!

L90: MR90-VPi and ROS-ITi17 (put here 'F0')

L91: resistance to: provide some examples of acaricides classes and provide reference(s).

L93: single male (put 'from F2' here)

Figure 1a legend: you need to indicate how recombination occurs for F2 (n) males.

L812-815: the formation of the individual populations should be better explained. Especially, the formation of the individual F2 (males) contributing to the formation of the populations 1-458.

L94: ...giving rise to an individual population (1-458 here, see Figure 1a).

L96-97: refer again to Figure 1a.

L97: per sample, or you mean per population, each of the 458? On average 42 per population sample?

L101-102: F0 for this cross, great-grand parents.

L112: please indicate the size of those bins. Information from a former study available?

Section genome-wide eQTL atlas: perhaps these numbers could be summarized in a table.

L124-127: important findings. Can you further discuss this in the appropriate section? See for example also: Kliebenstein D (2009) Quantitative Genomics: Analyzing Intraspecific Variation Using Global Gene Expression Polymorphisms or eQTLs. Annual Review of Plant Biology 60: 93-114.

Section trans-eQTL hotspots: please indicate how the trans-QTLs hotspots were distributed across the chromosomes.

L137-138: you do not mention here how many genes were affected by HS1.

L137-159: perhaps this section on GO terms could be streamlined and/or integrated into a table.

L154-155: remove 'for which significantly more target genes were upregulated than downregulated' as it is redundant information; already mentioned in L133.

L157: replace 'excepting for' with: except for

L159: HS5 has the most eQTLs associated, 10x more than HS2. Can you provide more information for HS5?

L162-163: describe how it was done.

L173: 'as well as in comparison to each other': is this redundant with information above (L170)?

L182: 'several CYPs annotated as pseudogenes in the reference assembly': this is interesting. Can you discuss this finding more within the appropriate section?

L203: 'CYP392A11 and its close homologs': which homologs? Is there a phylogenetic tree available?

L204: what do you mean by outsized importance?

L227: epistasis is usually defined as the interaction between genes. Remark: the CYP392A12 locus seems to show an additive allelic effect (low-intermediate-high: SS-RS-RR).

L255-295: I feel this section could be streamlined (shortened).

L303: replace 'outstanding' with: missing

L304: it is unclear what you mean exactly by environment. Please revise.

L304: causal genes of trans-regulation? Unclear.

L305: remove hyphen.

L308, 312: briefly explain here what and where on the genome (chromosomes) are HS1, HS5 and HS7. Indicate the underlying genes located at all three hotspots.

L310: what could be the indication for impact on fitness? Any genes important for primary metabolism found among the 1125 genes associated with HS5?

L315: please mention the number of associated detoxification genes in brackets.

L314-318: Both sentences could be merged and simplified.

L318: I think you should remind the reader here how many were in common (% of the trans-eQTL and % of the DEG results, respectively).

L320: CYP392A12 is an interesting candidate gene; Can you provide more information about its function?

L321: where there interactions (epistasis) between genes accumulating at different trans-eQTL hotspots? That would be interesting to mention here too.

L320-321: Indeed, there should be interaction between cis- and -trans variants. Please provide examples for this in insects, and/or in other organisms (plants, etc.), see for example: Potokina E, Druka A, Luo Z, Wise R, Waugh R, Kearsey M. Gene expression quantitative trait locus analysis of 16 000 barley genes reveals a complex pattern of genome-wide transcriptional regulation. Plant J. 2008 Jan;53(1):90-101. doi: 10.1111/j.1365-3113.2007.03315.x

L321: where there interactions (epistasis) between genes accumulating at different trans-eQTL hotspots? That would be interesting to mention here too.

L322: in which and/or in how many species specifically were those functional studies performed?

L325: can you briefly mention if you also found members of the AhR and CnC:Maf families in your dataset and what eQTL mapping or DEG analyses results showed for them?

L326: you should briefly introduce LBDs and DBDs here.

L327-331: more background on the functionality of LBD should be provided. Ditto information about functionality of DBD.

L332-333: I think a bit more in-depth information should be provided at this point.

How far apart are the HR96-1a and -1b genes mapping, are they in tandem? The RNAi results should be discussed in more depth here in the given context.

L334: IMPORTANT: it should be possible to infer the hierarchical order of regulation with your expression and mapping data.

L336: plant-produced aromatic compounds: such as? Please provide concrete examples!
L337-338: this should be a separate sentence, while providing more points of discussion for those results.
L343: diverse acaricides such as?
L343-345: this information should be found in the Materials and Methods section.
L348-353: could be streamlined. Also, it is not clear here to which work you are referring to. Snoeck et al.?
L348: which cytochrome P450 reductase? This is a multi-gene family.
L352: 'potentially the other acaricides to which the R strains is highly resistant': which other?
L353: sorry, I don't understand this reasoning.
L354-370: this section can be streamlined. Also, it seems largely speculative.
L372-375: this is a very interesting result, but it should be discussed more in depth.
L376-382: content seems a bit repetitive and redundant.

Reviewer #3:

Remarks to the Author:

The manuscript by Ji et al. describe elegant experiments that lead to substantial results. eQTL mapping and consequential genetic experiments, exhibiting a beautiful reductionist logic, identify a major trans-regulator of detoxification genes in the spider mite. The significance is multi-faceted; (i) there is new insights into a chemical resistance in a major pest, (ii) here is a new and rare example of insecticide resistance mediated by a trans factor, (iii) that resistance relates to gene copy number variation and divergence between those copies, (iv) these copies encode a nuclear hormone receptor that has ligand binding but not DNA binding domains, (v) here's a molecularly defined example of epistatic interactions relevant to a newly evolved fitness trait (vi) and enticingly, could this help explain dramatic changes in transcriptomes and general vigour of arthropods that switch host plants?

The eQTL experiment cleverly exploited the haplodiploidy of the tiny mites to generate 458 recombinant inbred families. RNA-Seq from the families not only yielded the transcript abundance traits that were mapped but also the SNPs that allowed mapping (prior genome sequencing of the parental strains was used as a reference). One trans-eQTL then became the focus of the paper because it upregulated multiple detoxifying genes, including those associated with insecticide resistance. The QTL was validated in two ways. Firstly, by independently introgressing the region into a common background to generate Near Isogenic Lines, that were then transcriptionally profiled. Secondly, the trans QTL was then validated by RNAi against a candidate gene – a nuclear hormone receptor gene (assessed by further transcript profiling). A nice experiment that arose from an exploration of the data was a test for epistasis between trans eQTL and a cis eQTL. The methodology seems really solid and I can see no flaws.

The following suggestions are mere trinkets on christmas tree:

1. All the variation considered existed between the two progenitor lines. We are told they differ in insecticide resistance profile, but the paper would be enriched by knowing more about the provenance of the lines. How many generations had they been in the lab? Were they collected off the same host plant? What host plant had they been reared on? Do they come from similar geographic regions? Should we think of these mites as panmictic?
2. Fig1a helped me understand the lines 92-97. The text (and my little-bit-of-knowledge-that-is-too-much) didn't reveal that sons inherit recombined chromosomes (it now seems obvious that recombination occurs in the eggs that are unfertilised although pseudo-arrhenotoky does exist: so I am not recommending a change here just noting a path I went down). More importantly, I wonder whether the word 'samples' (I92 and I99) is better replaced with something more precise (I used 'families' above).
3. The b copy of the nuclear hormone gene HR96-LBD-1 gene differs between the resistant and susceptible strain by a radical amino acid substitution W309R. Is it worth inspecting this using alphafold structure predictions? Even if you look at the homologous region in the Drosophila version which is available through flybase would probably add more to supp figure 8.
4. Is it worth looking for common motifs in each gene that is regulated by transQTL 1? Would such

analysis explain why some genes are more greatly upregulated?

5. It would be an even better paper if there was evidence that the trans-QTL showed signs of being recently selected. Could a junction spanning PCR trivially assess how common the CNV is at HR96-LBD-1? Are there signs of selective sweeps at the locus?

6. Because time is needed to evolve sophisticated regulatory responses, induction may be more important for 'plant chemical challenges' than 'pesticide resistance'. It is very exciting that you now associate transregulatory changes with the latter. But I feel that there is something more general that can be made of the former: what do these results say about host switching responses in arthropods more generally (anything to say to those studying whiteflies and aphids)?

AUTHOR RESPONSES TO REVIEWERS

Note: We have now plotted all data points for barplots and boxplots ($n \leq 10$), and made minor changes to the legends to explicitly indicate what was plotted (means, etc.). These minor changes were to Fig. 3e, Fig. 4d, Fig. 5b, and Supplementary Figure 10, the respective legends, as well as the legends for Supplementary Figures 4 and 5. We also fixed one small plotting issue with Fig. 4d.

Just to be clear, this was done to ensure that the plots now conform to the required Nature Communication guidelines for figures; the respective source data, analyses, and conclusions are not affected.

REVIEWER COMMENTS

Reviewer #1 (Remarks to the Author):

This is a very interesting study that leverages some of the unique aspects of the *Tetranychus urticae* system to obtain significant new results on trans-acting factors regulating families of detoxicative genes in this agricultural pest. The work started with two different inbred strains, one with high constitutive expression of many putative detoxicative genes. A crossing design produced haploid F2 males carrying recombinant chromosomes, which were individually crossed to females of the other strain, producing identical F3 daughter cohorts to generate enough tissue for RNA-seq analysis. Then the males (or their daughter cohorts) were genotyped to identify recombinant chromosomal regions that influenced differentially-expressed genes identified by RNA-seq. cis- or trans-acting factors could thus be identified by comparing the chromosomal region with the location of the differentially expressed gene. One especially influential trans-acting region was investigated in more detail, and found to contain a tandem duplication of an HR96 gene. RNA interference of both duplicated genes had effects on target detoxicative genes consistent with the eQTL analysis.

The system is very impressive and the results are quite rich and complicated. Although the results do not reveal a novel family of transcriptional regulators, they are important in establishing the universality of the HR96 genes. However, from a mechanistic point of view, the results are frustratingly incomplete. Since the identified HR96 genes lack DNA-binding domains, indirect effects due to dimerization with other unknown family members with DNA binding domains must be hypothesized. Since it was not possible to target each duplicate HR96 gene separately with RNAi, the significance of the "radical tryptophan to arginine" substitution is not resolved. Perhaps the authors could check to see whether this residue occupies a site in the dimerization domain--this would not require additional experiments. This ambiguity also hampers understanding of the very interesting epistatic interaction between the trans-acting transcription factor and the cis-acting regulatory sequence. Perhaps the authors could probe the difference between the R and S strains in the cis-factor--this would not require additional experiments. A more complete mechanistic understanding would require different approaches; the authors have pushed the genetic approach as far as it can go. This should not stand in the way of publication, as the study nicely illustrates the limitations as well as the strengths of the genetic approach in the fascinating and economically important *Tetranychus* system.

The manuscript should be rewritten for a general audience. The style is dense and highly compressed, requiring that a sentence be read several times to unpack the meaning. The frequent use of abbreviations and acronyms is convenient for the authors but not for the readers. Each paragraph in the Results would benefit from an introductory sentence motivating the results, instead of leaving it to the reader to deduce

the motivations after having read the entire paragraph.

[Please note: A document with the changes made in response to all reviewers is provided with the changes shown with the track change feature. Please also note that we added fold change information to what was originally Supplementary Data 8, but is now Supplementary Data 11 in the resubmission. This makes it easier to compare fold changes across the experiments performed in the study.]

First, we appreciate the thoughtful and helpful comments. Reviewer 1 raises three points.

1. Writing style.

Our original draft of the manuscript was substantially longer (and perhaps clearer in some respects) than the original submission, but we reduced it to approximate the suggested length for the journal. The suggested length for a Nature Communications article is 5000 words for the Introduction, Results, and Discussion, and our original submission was 4998 words. A result, partly out of necessity, was a very dense manuscript given the scope of the studies that we performed.

However, after consulting with the editor, we have lengthened the revised manuscript as there is no way around doing so to address the comments/concerns of the three Reviewers (and there is little that can be cut given how dense it was originally). With respect to writing style, for the beginnings of sections, or paragraphs within sections, we have modified the text to attempt to give a gentler and more structured introduction to what follows. Also, throughout we have edited the manuscript in an attempt to clarify sentences.

Examples of this include the beginning of the following Results sections: “Dense recombination in *T. urticae*”, “Genome-wide eQTL atlas”, “A trans-eQTL hotspot controls expression of many detoxification genes”, and “Characterization of HS1.”

2. Analysis of the radical amino acid change.

With respect to the W309R change, Reviewer 1 notes “Perhaps the authors could check to see whether this residue occupies a site in the dimerization domain--this would not require additional experiments.”

This is an excellent question, and this was also brought up by Reviewer 3 who commented on this more at length and suggested using alphafold-based modeling to investigate the change. We have now done this, and the result provides additional clarification about what the change might do (it is predicted to be inward facing in the ligand-binding pocket itself, which is striking). Please see our detailed response to Reviewer 3 point 3, a new Supplementary Figure in this revision (Supplementary Figure 11), and the respective changes to the Results and Discussion (again, please see responses to Reviewer 3 point 3).

3. Examination of *cis* variants:

Reviewer 1 notes that “Perhaps the authors could probe the difference between the R and S strains in the *cis*-factor--this would not require additional experiments.”

We are also very interested in understanding the result with CYP392A12 (we hope we understand correctly that this is what the Reviewer’s comment refers to). However, it has been known since the first publications of variant data for *T. urticae* strains that polymorphisms are ubiquitous in this species (about one SNP every 100 bp on top of substantial indel variation; i.e., see Grbic et al., 2011. Nature 479:487-492, PMID: 22113690, Van Leeuwen et al., 2012. PNAS 109(12):4407-4412, PMID: 22393009). Trying to associate variation in sequences around promoters to effects on expression is therefore challenging in

T. urticae, and for that matter remains so in most organisms (even for human and other vertebrates, which can have more than an order of magnitude fewer sequence differences between two individuals than observed between *T. urticae* strains). Also, very little is known in *T. urticae* about the distances over which regulatory elements act, further complicating attempts to ascribe differences to specific DNA polymorphisms). While there are many differences between the R and S strains in and around CYP392A12 (which is expected), we believe expanding the paper to look at this, and by analogy other instances with other genes, would just lead to a level of conjecture that we are not comfortable with. Even in *Drosophila melanogaster* where promoter bashing to narrow down differences is possible, it is still not straightforward and there is no guarantee of success. We hope the tools to perform these types of analyses become possible in future, but they are not yet there in *T. urticae* (or for almost any arthropod outside of *D. melanogaster*, in fact, including even other *Drosophila* species).

Reviewer #2 (Remarks to the Author):

Review of Ji et al. on the identification of a gene underlying large trans-effect upregulation of detoxification and host plant use genes in a generalist herbivore.

The authors created 458 F3 backcross females, which were subsequently used in eQTL mapping, by generating three generation crosses of *Tetranychus urticae*, by firstly mating an acaricides resistant with a susceptible strain; the resulting F1 females were kept unmated to produce hybrid recombinant sons (F2); then, these haploid F2 males were backcrossed into females from the susceptible strain, which ultimately generated the diploid F3 backcross females. Information on genotype data (using 1889 informative genotype bins) with information on gene expression phenotypes were combined to identify (using a linear model) 5,685 local associations (cis-eQTLs, with 54.7% within +/- 100kb) and 10,563 gene expression associations in trans (trans-eQTLs). A total of 9,740 genes were found associated with eQTLs when exclusively considering those genes located on the chromosome assembly of the reference genome of *T. urticae*. A considerable number (537) of genes with eQTLs were identified that are of interest for specialized metabolism, transport and detoxification in explaining resistance and host use. Several eQTL hotspots with an accumulation of trans-eQTLs were identified, of which, a hotspot termed HS1 was further investigated. To do this, near-isogenic lines were created. The resistance haplotype at hotspot HS1 was introgressed into the susceptible genetic background. Then, the authors searched for an overlap between differentially expressed genes between those lines and the trans-regulated genes at HS1 from the eQTL study. Resistance was also found related to higher expression at the CYP392A12 locus when the resistance allele was present (this finding was validated by introgressing this locus from the resistant into the susceptible *T. urticae* strain to create an RS genotype and comparing expression levels for this locus with the parental strains, SS or RR at that locus). Further investigation of the HS1 was done by exploring the genomic region underlying this hotspot. To do this, long-read sequencing was used to improve this genomic region's assembly, where HR96-like genes were identified to be in fact tandemly repeated. The influence of those two tandemly repeated genes on the trans-regulation of genes by HS1 was checked by jointly knocking both loci down in an RNAi experiment, and studying resultant transgenic *T. urticae* lines for their performance on bean and tomato hosts, with tomato the more challenging host for this generalist herbivore. Many detoxification genes that were otherwise upregulated in resistant (RR-HS1) versus susceptible (SS-HS1) near-isogenic lines were downregulated in RNAi background of HR96-like genes (those previously found within HS1 location from the eQTL analysis).

In sum, the authors presented a very interesting work related to an important topic that is the biology and evolution of herbivory. They used genetical genomics to infer important regions on the genome of a generalist, regions that are potentially related to the herbivore's host plant use potential. Several lines of evidence suggested that important genetic determinants of host plant use metabolism were unravelled. However, two points could be added to the analyses: 1) integrate phenotypic data that define acaricide resistance and identify collocation with eQTLs, and thus, improve information about genes that directly

underlie this specific resistance trait; 2) an analysis of the hierarchical order of gene regulation based on trans-eQTLs for each gene to deduce a 'global regulatory hierarchy' in resistance. Also, I found the Discussion limited (with sometimes redundant information provided); it could be improved by providing more in-depth insights for certain points (comments provided below).

[Please note: A document with the changes made in response to all reviewers is provided with the changes shown with the track change feature. Please also note that we added fold change information to what was originally Supplementary Data 8, but is now Supplementary Data 11 in the resubmission. This makes it easier to compare fold changes across the experiments performed in the study.]

First, we appreciate the thoughtful, detailed and helpful comments. Reviewer 2 raises two major points.

1. “Integrate phenotypic data that define acaricide resistance and identify collocation with eQTLs, and thus, improve information about genes that directly underlie this specific resistance trait.”

Our laboratories have been at the forefront of high-resolution bulked segregant analysis (BSA) QTL mapping of acaricide resistance phenotypes in *T. urticae*. While there remain relatively few of these studies, we have mapped a resistance QTL to the *HR96-LBD-1a* and *HR96-LBD-1b* genes in a very closely related strain to the R strain. The QTL was for resistance to the mitochondrial electron transport inhibitors of complex I (METI-Is) acaricide tebufenpyrad. We mentioned this in the original submission Discussion (and this is retained in the resubmission); however, we did not clearly report that the parental population from which the R strain (MR-VPi) was inbred, the MR-VP population, was selected long-term for high resistance to tebufenpyrad (this is now explicitly described in the Methods section “Mite strains and husbandry”). See also response to Reviewer 3 point 1.

Other peaks in the existing acaricide QTL mapping data for *T. urticae* do not overlap *HR96-LBD-1a* and *HR96-LBD-1b* (at least not significantly), but were mostly done with unrelated strains; hence, we do not know if the relevant variation we see between the R and S strains in the current study would have been segregating in those other crosses (i.e., Wybouw et al., 2019. *Genetics* 211(4):1409–1427, PMID: 30745439, Fotoukkaia et al., 2021 *PLOS Genet* 17(6):e1009422; PMID: 34153029, De Beer et al., 2022. *Insect Biochem Mol Biol* 145: 103757; PMID: 35301092, De Beer et al., 2022. *Biology* 11:1630; PMID: 36358331, Villacis-Perez et al., 2023. *Evol Appl* 00:1–17; PMID: 37124092).

It is not clear if any of the other hotspots besides HS1, with the possible exception of HS6 and HS7 (see Results), are related to the control of detoxification gene expression. We want to be very clear that any two strains are expected to have extensive expression variation, and hotspots, regardless of their resistance status (in any cross, strains of whatever organism vary in many traits, not just the ones investigators are targeting for study). So, it is unclear if it makes sense to relate HS2-9 to resistance QTL at all, and the same caveats about different strains, etc., would apply.

In summary, while we appreciate the Reviewer’s sentiment, we believe that trying to broadly link the phenotypes at specific genes that have eQTLs, whether from *trans* or *cis* effects, to resistance phenotypes inferred from other strains (and often with other acaricide compounds) would introduce substantial conjecture and length to the manuscript at the expense of clearly explaining the study’s key question and the major findings. We do focus in more detail on some detoxification genes regulated by *HR96-LBD-1a/HR96-LBD-1b* in *trans*, especially CYP392As. However, this situation is different because there is functional evidence from other studies that those specific genes may be involved in metabolizing some of the exact acaricides to which the R strain is resistant.

2. “an analysis of the hierarchical order of gene regulation based on trans-eQTLs for each gene to deduce a 'global regulatory hierarchy' in resistance.”

We can understand why the Reviewer wishes for this analysis to be performed (it would be great if it could be accomplished). Our study began with two strains that we knew varied greatly in acaricide resistance owing to very large *trans*-driven genetic effects (Kurlovs et al., 2022. PLOS Genet 18(11):e1010333; PMID:36374836) These large differences turned out to be due overwhelmingly to the effect of the HR96-LBD-1a/HR96-LBD-1b master regulator at HS1. The effects on expression of the other detoxification genes in the genome from other *trans* eQTL are minor in magnitude by comparison, and it is the large-effect differences in detoxification gene expression that have typically been associated with resistance phenotypes in arthropods (i.e., Nauen et al., 2022. Annu Rev Entomol 7:105-124; PMID: 34590892). Further, HR96 proteins, and also CncC and AhR genes that Reviewer 2 mentions as well (see below), are regulated primarily by sensing xenobiotics post-transcriptionally. Therefore, our expression data is likely not suitable to detect the type of “global regulatory hierarchy in resistance” that the Reviewer asks, and to attempt it would likely give a very distorted view of a hierarchy as some of the key regulatory mechanisms would likely be missed entirely. And, if one does “walk” from one eQTL to the other, there are still so many candidate genes in the regions that trying to construct a hierarchical order tied to specific genes would we fear be highly subjective (and any inferences would potentially be very limited to just the R and S strains). Honestly, in our study, we were very fortunate to have the findings we did for HS1 (but that is what the data shows, and it is surprising beyond anything we expected going into the study).

We have taken Reviewer 2’s major comments seriously, and have implemented many of the minor suggestions as described below (thanks for taking so much time with the manuscript). However, we think that addressing points 1 and 2 above is complicated or confounded for the reasons we mention, and attempts to do so would come with many caveats that would need substantial additional space to be conveyed appropriately, and in that sense would distract from the key focus and findings of our study that are straightforward and have very strong experimental support.

Other comments:

L25: please specify these additional approaches (reverse genetics?)

The sentence now reads: “As established by additional genetic approaches including RNAi gene knockdown, a duplicated gene...” The other approaches also include the NILs, etc., but we think this captures it better and we are at the abstract word limit.

L56-57: what is the chemical basis of these pesticides?

The chemical bases are diverse, and are reviewed in the reference that is cited in the sentence. However, with respect to the S and R strains, we now provide information about the specific compounds to which they are resistant and the respective pesticide classes. Please see revisions to the Methods section “Mite strains and husbandry”. See also response to Reviewer 3 point 1.

L65: name a few of such genes here

There are hundreds of such genes that change in expression, and this is covered in the references that are provided. We believe that providing the names of a few specific genes is potentially distracting from the bigger picture. And, by virtue of providing specific names, it might influence the readership into believing that only a small number of genes, or only genes in a few families, responded (which is not true).

L74: where do these lines originate from?

We acknowledge that we should have provided more information on these strains in general. This information, including details about acaricide exposure, dates of collection, etc., is now provided in the Methods section “Mite strains and husbandry”. See also response to Reviewer 3 point 1.

L79-80: more specific, numbers?

We performed two large RNAi experiments, and the numbers do differ on bean and tomato (but in an informative way). But, condensing this down to the last summary paragraph of the Introduction is fraught with the opportunity for misunderstanding, so we think this is best presented in the Results where the specific experiments are clearly elaborated (that is, the knockdowns on bean and tomato).

L79: remove 'the' before 'most'.

Thanks for catching this mistake. Done.

L80-81: are you talking here about copy number variation since they are tandemly duplicated, see L78/9?

We have replaced “respective” with duplicate to clarify what we meant (thanks for pointing out how our wording could have been confusing).

L83: 'intra-specific variation in regulators': this should be more specific!

The sentence now reads: “Therefore, segregating genetic variation in regulators of xenobiotic pathways can be a source of the dramatic upregulation of detoxification genes associated with pesticide resistance evolution and host plant adaptation in arthropod species.”

L90: MR90-VPi and ROS-ITi17 (put here 'F0')

We have added “F0 generation”.

L91: resistance to: provide some examples of acaricides classes and provide reference(s).

We worry that for the general readership of Nat. Comm., getting too down into the details about the specific acaricides at this point in the manuscript might be distracting. But, we have modified the respective sentence to say how many compounds and classes, and we refer the reader to the revised Methods (Methods section “Mite strains and husbandry”). See also the response to Reviewer 3 point 1.

L93: single male (put 'from F2' here)

We have substantially rewritten this paragraph (first paragraph in Results section “Dense recombination in *T. urticae*”) to clarify the experimental design. Please see track changes. These changes include clearly indicating that the single males are the F2 generation. Please see also our response to Reviewer 3 point 2.

Figure 1a legend: you need to indicate how recombination occurs for F2 (n) males.

L812-815: the formation of the individual populations should be better explained. Especially, the formation of the individual F2 (males) contributing to the formation of the populations 1-458.

We have substantially rewritten the relevant parts of the Figure 1 legend to clarify what was done; please also see the response to Reviewer 3 point 2. We believe the changes to the main text and figure legend dramatically improve the description of the experimental set up.

L94: ...giving rise to an individual population (1-458 here, see Figure 1a).

We have substantially rewritten this paragraph (first paragraph in Results section “Dense recombination in *T. urticae*”) to clarify the experimental design. Please see track changes. Please see also our response to Reviewer 3 point 2.

L96-97: refer again to Figure 1a.

We have substantially rewritten this paragraph (first paragraph in Results section “Dense recombination in *T. urticae*”) to clarify the experimental design. Please see track changes. Please see also our response to Reviewer 3 point 2.

L97: per sample, or you mean per population, each of the 458? On average 42 per population sample?

We have substantially rewritten this paragraph (first paragraph in Results section “Dense recombination in *T. urticae*”) to clarify the experimental design. Please see track changes. Please see also our response to Reviewer 3 point 2.

L101-102: F0 for this cross, great-grand parents.

We have added F0.

L112: please indicate the size of those bins. Information from a former study available?

We have added this to the sentence to give some context (“..... 1889 maximally informative genotype bins (median length of 29.5 kb) based on observed recombination events”). But, there has been a misunderstanding. The bins were not from an earlier study, but rather were generated in this study using prior DNA-seq data and the RNA-seq data for the 458 F3 families generated as part of this study – specifically, they were calculated as shown in Supplementary Figure 2 (a schematic to illustrate bin assignment) using the recombination data presented in Supplementary Data 1. Please see also the beginning of the Methods section “eQTL mapping.”

We have also now added a Supplementary Data file (Supplementary Data 2) with this information as part of this resubmission. Now the information about every marker bin is easily accessible to the readership.

Section genome-wide eQTL atlas: perhaps these numbers could be summarized in a table.

We did consider a table, but in reality even if there were a table (which would add one more display item to an already long manuscript), we still think we would need to devote about the same amount of text to guide the reader through these results. So, in the end, we opted not to have a main text table (but see, for instance, Supplementary Data 3).

L124-127: important findings. Can you further discuss this in the appropriate section? See for example also: Kliebenstein D (2009) Quantitative Genomics: Analyzing Intraspecific Variation Using Global Gene Expression Polymorphisms or eQTLs. Annual Review of Plant Biology 60: 93–114.

We have changed the sentence to read: “For all genes, as well as for the detoxification genes, $-\log_{10}$ (adj-p) values for *cis* eQTL were significantly greater than for *trans* eQTL; similar trends were observed when examining absolute values of effects sizes (beta values from a linear model; Wilcoxon rank sum tests, all

p < 10⁻¹⁵; Supplementary Fig. 4), a finding observed in related studies in other animals and plants (²⁸ and references therein).”

This makes it clear that the findings for *cis* eQTLs are not unexpected (greater effect sizes on average than for *trans* associations), and the readership is referred to the review by Hill et al., 2021. Nat Rev Genet 22:203–215; PMID: 3268840 for a deeper understanding (ref 28, which is a more recent review than the Kliebenstein 2009 reference, which admittedly would also be a completely fine reference to use). We don't discuss this finding for *cis* eQTL at length just because it has been observed in many studies and the focus of our paper is on *trans* control of detoxification gene expression (and we are space limited, please also see response to Reviewer 1 point 3).

Section trans-eQTL hotspots: please indicate how the trans-QTLs hotspots were distributed across the chromosomes.

This comment, and the ones below by Reviewer 3, have brought to our attention that we left out some information that ideally should have been in this section. In response, we have rewritten much of the first paragraph of the section “A *trans*-eQTL hotspot controls expression of many detoxification genes”, and made other changes throughout this section, please see track changes.

We now indicate the distribution of hotspots across the chromosomes (the reader does not have to dig this information out from Fig. 2c). Also, we indicate in the text the location of hotspots that we focus on (the genomic coordinates, and the number of *trans*-regulated genes). Further, we have added a Supplementary Data file (Supplementary Data 7) in the resubmission that has the genes located at each of HS1-9, and the coordinates for all the hotspots.

Additionally, we also now discuss HS6 in this section specifically (and now HS6 is also briefly mentioned in the Discussion, second paragraph). HS6 had only one enriched GO term and it was only marginally significant and supported by only a small number of genes. But, the term was detoxification related (CYPs), and for thoroughness we have now added specific information about HS6 to this section.

L137-138: you do not mention here how many genes were affected by HS1.

We have now added this (“In a gene ontology (GO, based on molecular function) enrichment analysis with genes controlled in *trans* by HS1 (chromosome 1, 12.4-12.6 Mb, 182 genes),”). Please also see Supplementary Data 6 in the resubmission that has this for all hotspots and that we retained from the original submission.

L137-159: perhaps this section on GO terms could be streamlined and/or integrated into a table.

We considered this seriously in writing the original manuscript, but in the end thought that a text description of these findings might be best to relay the information (and a table would have been one more main text display item). And, this is so important to convey for HS1, as only HS1 had clear and dramatic (highly significant) enrichments for detoxification genes controlled in *trans*.

L154-155: remove 'for which significantly more target genes were upregulated than downregulated' as it is redundant information; already mentioned in L133.

Thanks, fixed.

L157: replace 'excepting for' with: except for

Thanks, fixed.

L159: HS5 has the most eQTLs associated, 10x more than HS2. Can you provide more information for HS5?

In the original submission, as well as in the resubmission, we note that one GO term is enriched, and that this hotspot is coincident with a region of genotypic ratio distortion in the cross. This indicated that this genomic region likely impacts relative fitness, and it would not be surprising if any deleterious allele in one strain versus the other has an impact on gene expression (potentially for many genes, albeit note that significance values and effect sizes for genes impacted by HS5 in *trans* are minor despite the large number). Of course, all strain and individual genomes have such loci (several tens of potentially lethal deleterious alleles are present in any given human in the heterozygous state, for example). So, finding hotspots with this behavior is not, in our, opinion, unexpected.

However, saying what this hotspot “does” is not straightforward, and it is not clear what more we can say at this point. And, for this paper, which focuses on understanding large *trans*-driven differences in detoxification gene expression, no GO terms associated with detoxification genes were enriched. We only saw marked enrichment for detoxification terms for the *trans*-regulated HS1 gene set – that is why we transition immediately after this section to focus on HS1.

L162-163: describe how it was done.

As described in the sentence, the NILs were constructed by marker-assisted backcrossing. This was done with PCR and CAPs markers, as described in the Methods section “Construction and characterization of NILs for HS1.” The primers used, and the restriction enzymes, are provided in Supplementary Data 19 (as numbered in the resubmission). We now refer to the Methods explicitly from this sentence. The revised sentence now reads: “To do this, we first constructed two independent sets of near-isogenic lines (NILs) by marker-assisted backcrossing in which the R haplotype at HS1 was introgressed into the S genetic background (see Methods and Supplementary Fig. 7).”

L173: 'as well as in comparison to each other': is this redundant with information above (L170)?

This can be removed and has been.

L182: 'several CYPs annotated as pseudogenes in the reference assembly': this is interesting. Can you discuss this finding more within the appropriate section?

Our feeling is that this may or may not be that interesting just because *T. urticae* strains are known to be highly polymorphic, and what may be annotated as a pseudogene or fragment in one strain (in this case the London reference strain) may or may not be in other strains. And, there are many instances in which it is known that pseudogenes are expressed and may retain ancestral control for long periods of time (even if the product has mutations that disrupt the coding sequence). Because of this, we have viewed the findings with pseudogenes as not, potentially, distinct from what is observed for other genes (at least in the context of *trans* control of detoxification genes, the focus of our study).

L203: 'CYP392A11 and its close homologs': which homologs? Is there a phylogenetic tree available?

CYP392A11 and CYP392A16 specifically. Members of these closely related CYPs have been functionally characterized and shown to metabolize several acaricides (please see the Discussion). And, yes, for our statement “This gene is the most similar CYP to CYP392A11 in *T. urticae*²⁰”, the reference

we cite is for the original *T. urticae* genome paper, which has a phylogeny of the *T. urticae* CYPs (see Fig. 2a, CYP2 clan, in that paper: Grbic et al., 2011. Nature 479:487-492, PMID: 22113690).

L204: what do you mean by outsized importance?

This has been changed to read "...*CYP392A11* and its close homologs in *T. urticae* appear to be important in the development of acaricide resistance..." for clarity.

L227: epistasis is usually defined as the interaction between genes. Remark: the *CYP392A12* locus seems to show an additive allelic effect (low-intermediate-high: SS-RS-RR).

Yes but in terms of the high-level expression of this gene there is a very large jump in expression level when the genotypes are RS at both loci as compared to all the other combinations.

L255-295: I feel this section could be streamlined (shortened).

We have attempted to shorten and clarify this section (including moving up the GO results and removing the last paragraph). This shortens the section by a little, but obviously this is a truly critical section in the Results so we were careful not to remove critical information.

L303: replace 'outstanding' with: missing

We have replaced "outstanding" with "remain unanswered."

L304: it is unclear what you mean exactly by environment. Please revise.

The sentence now reads: "In particular, what are the loci underlying *trans*-mediated variation in gene expression, and are they the same or different from those in metazoan xenobiotic response pathways established by molecular genetic studies?"

L304: causal genes of trans-regulation? Unclear.

The sentence now reads: "In particular, what are the loci underlying *trans*-mediated variation in gene expression, and are they the same or different from those in metazoan xenobiotic response pathways established by molecular genetic studies?"

L305: remove hyphen.

Done.

L308, 312: briefly explain here what and where on the genome (chromosomes) are HS1, HS5 and HS7. Indicate the underlying genes located at all three hotspots.

In the revised results section, we have now very clearly indicated where in the genome the hotspots are located, and the number of genes they regulate in *trans* and the number of genes within the hotspots (see Supplementary Data 6 and Supplementary Data 7). Please see above. Because we have done this, and the information is there in the figures and supplementary materials (original submission and resubmission), it could be debated whether this needs to be done again in the Discussion.

However, we defer to the Reviewer and have revised the second paragraph of the Discussion to have the location (chromosome and position) and number of *trans*-regulated genes for the hotspots that we mention by name.

L310: what could be the indication for impact on fitness? Any genes important for primary metabolism found among the 1125 genes associated with HS5?

See response above on HS5 (Reviewer query about L159).

Also, we have changed the sentence to make it clearer how genotypic ratio distortion could be linked to a locus impacting fitness. The sentence now reads:

“Our finding of more *trans* than *cis* eQTL for all genes, as well as for detoxification genes, is explained in part by eQTL hotspots, such as HS5 on chromosome 2 (2.6-3.2 Mb, 1125 *trans*-regulated genes) that was also coincident with genotype ratio distortion in the eQTL mapping population, a signal of a segregating variant that impacts fitness (i.e., one that differentially impacts survival, as can be detected in multi-generational experimental designs).”

L315: please mention the number of associated detoxification genes in brackets.

We are somewhat conflicted about adding this to the Discussion, as a strong argument could be made that this belongs in the Results section (where this information is covered). Nevertheless, we have modified the sentence to read:

“..... Among these three hotspots, HS1 was exceptional in both the percentage of detoxification genes with associations (30.8% versus 8.1% and 13.7% for HS6 and HS7, respectively, Supplementary Data 6), and in the magnitude of”

This does, however, reduce it to numbers, and the difference is striking, but equally important, if not more so, is that many of the detoxification genes *trans*-regulated by HS1 have very large differences in expression (what is covered in the second half of the sentence).

Also, we are a little conflicted as well to include this because the 30.8% only includes known and annotated “detoxification genes” even though some of the other genes are, for example, the cysteine proteases that are associated with host plant adaptation in *T. urticae*. Regardless, this is covered in the Results.

L314-318: Both sentences could be merged and simplified.

The experiment with the NILs uncouples *cis* effects and also *trans* effects coming from other loci in the genome in the eQTL mapping population. Therefore, we believe that the findings for the NIL work should be covered in a separate, dedicated sentence, as the two analyses are different if complementary (reassuringly, of course, they gave strikingly similar results).

L318: I think you should remind the reader here how many were in common (% of the *trans*-eQTL and % of the DEG results, respectively).

Of all the genes that change in expression, we are most interested in the detoxification and host-plant-associated genes, and among the genes with large differences in fold change, the genes in common go up (this is completely expected based on a simple understanding of statistical power and effect sizes). That is, the number in common depends in part on effect sizes, as well as *cis* and *trans* interactions. We feel

that going into the level of detail in the Discussion that is needed to properly understand the comparison the Reviewer asks for is potentially distracting for the readership (and the information is in the Results and linked Supplementary materials).

But our statement about epistatic interactions as one source of difference is factually correct. A main purpose of the sentence was to raise the findings with *CYP392A12*, which we validated conclusively with substantial effort (Fig. 4).

L320: *CYP392A12* is an interesting candidate gene; Can you provide more information about its function?

In the original submission (and in the resubmission), we note:

Results section: "A trans-eQTL hotspot controls expression of many detoxification genes. "

"These included *CYP392A11* and *CYP392A12* that were previously shown to be highly expressed in the R strain compared to several acaricide susceptible *T. urticae* strains¹⁷."

Result section "An epistatic interaction underlies heightened expression of *CYP392A12* in the R strain":

"While the relevance of this observation is not clear, one of the 25 genes for which a *cis* effect was associated with upregulation by the RS genotype was *CYP392A12*. This gene is the most similar CYP to *CYP392A11* in *T. urticae*²⁰, and *CYP392A11* and its close homologs in *T. urticae* appear to be important in the development of acaricide resistance as demonstrated by recent genetic and functional studies^{9,30}. *CYP392A12* is located 27.4 kb from *CYP392A11* on distal chromosome 1 (Supplementary Data 5), and like *CYP392A11* has a *trans* association with higher expression associated with the RS genotype at HS1."

These sentences explain earlier findings, and cite references that demonstrate that *CYP392A12* is the most closely related CYP to *CYP392A11* (the most upregulated CYP by HS1/NILs) for which functional studies have been performed (they are also likely ancient tandem duplicates, see our comment about their genomic positions).

This is what is known about *CYP392A12* from prior work, but we think it is substantial (and we feel that rehashing this in the Discussion might distract the readership).

L321: where there interactions (epistasis) between genes accumulating at different trans-eQTL hotspots? That would be interesting to mention here too.

We agree with the Reviewer that this is interesting. However, this analysis would require substantial additional analyses that, we feel, would distract from the presentation of the major findings that we believe make our current work high impact (this type of analysis, including also non-detoxification and host-plant-associated genes at all the hotspots, is plausibly more suitable for a more specialized genetics journal).

L320-321: Indeed, there should be interaction between *cis*- and *-trans* variants. Please provide examples for this in insects, and/or in other organisms (plants, etc.), see for example: Potokina E, Druka A, Luo Z, Wise R, Waugh R, Kearsey M. Gene expression quantitative trait locus analysis of 16 000 barley genes reveals a complex pattern of genome-wide transcriptional regulation. *Plant J.* 2008 Jan;53(1):90-101. doi: 10.1111/j.1365-313X.2007.03315.x

Yes there should be, and this has been shown in many systems (so that is not surprising). But, we specifically discuss interactions for “detoxification” genes, and what we meant is arthropod detoxification genes (see the Introduction which gives the set up to the work; we have now clarified the sentence to end in “...highlighting the potential importance of interactions between *trans* and *cis* variants in the origin of expression variation in arthropod detoxification genes.”). Few studies have specifically investigated *cis* and *trans* interactions in detoxification genes in arthropods (really, it takes a study like ours to do that, and there are few, especially ones that specifically discuss detoxification genes). We therefore did not uncover any references specifically for such *cis-trans* interactions for detoxification genes in arthropods.

L321: where there interactions (epistasis) between genes accumulating at different trans-eQTL hotspots? That would be interesting to mention here too.

Please see our response to the question about L321 above (it appears that somehow this got duplicated).

L322: in which and/or in how many species specifically were those functional studies performed?

This sentence has been reduced in the resubmission because some of what was covered in the sentence is now moved to the Results section “Tandemly duplicated nuclear hormone receptor-96 (HR96) like genes at HS1.” (please see also response to query about L325 below).

However, we cited a recent review that covers all this in great detail (Amezian et al., 2021, Pestic Biochem Physiol; PMID: 33838715).

Please see also our response to Reviewer 1 (Point 1, writing style); in cases like this, we need to use citations for prior work else our manuscript risks becoming too lengthy which might be at the expense of a broad readership like that of Nature Communications.

L325: can you briefly mention if you also found members of the AhR and CncC:Maf families in your dataset and what eQTL mapping or DEG analyses results showed for them?

We acknowledge that we should have commented whether the CncC:Maf and AhR genes, and the many other HR96 and HR96-like genes in *T. urticae*, colocalized with other hotspots. It is true that CncC:Maf and AhR have generated intense interest in the field. However, this analysis really should be in the Results section.

Therefore, we have moved this sentence (with modification) to the Results section “Tandemly duplicated nuclear hormone receptor-96 (HR96) like genes at HS1.” Please see the rewritten first paragraph. We now note in this section that no CncC:Maf or AhR genes are present at HS1 or in the other hotspot regions. We did find, however, that two other *T. urticae* HR96 genes are present in the intervals for HS8 and HS9 (see also Supplementary Data 7 in the resubmission). However, these two hotspots are not enriched for genes associated with detoxification, so it has to just be an observation at this point (and as there are 55 of these genes in the *T. urticae* genome, it might just be coincidence).

Supporting this analysis, we have added a new Methods section “Detoxification genes and their potential transcriptional regulators” that describes the source/annotation of these genes (for instance, we had to annotate AhR in *T. urticae* for this analysis as it was not done previously).

Also, we did not observe differential gene expression of any of the *T. urticae* CncC homologs or the AhR ortholog in the RR versus SS NIL DEG analyses. And, we did not observe differential expression for these genes, or NHRs, other than *HR96-LBD-1a* and *HR96-LBD-1b*, in the treatment versus control RNAi experiments (this latter point was noted in the original submission, and remains in the resubmission).

L326: you should briefly introduce LBDs and DBDs here.

We have now given more information about LBDs and DBDs when they are first introduced in Results section “Tandemly duplicated nuclear hormone receptor-96 (HR96) like genes at HS1”. These are now introduced in more detail at the first point they are mentioned in the manuscript (in the Results section, because the reader needs to understand at that point in the manuscript as in the Results section we discuss that HR96-LBD-1 only has the LBD). Please see also our response to L325 above.

L327-331: more background on the functionality of LBD should be provided. Ditto information about functionality of DBD.

Please see our response to L326 above.

L332-333: I think a bit more in-depth information should be provided at this point. How far apart are the HR96-1a and -1b genes mapping, are they in tandem? The RNAi results should be discussed in more depth here in the given context.

As explained in the Results “Tandemly duplicated nuclear hormone receptor-96 (HR96) like genes at HS1”, and as was shown in Fig. 5a in the original submission and in the resubmission, the genes are in tandem. Regardless, we could have provided more information about the exact distances, and a detailed schematic of the *HR96-LBD-1a/b* genes in the R, S and C1N1d strains is now provided in a new supplementary figure (Supplementary Figure 8; please note that both genes are deleted in C1N1d, please see response to Reviewer 3 point 5).

With respect to the RNAi results, they are presented in great detail in the Results section “RNAi knockdown of *HR96-LBD-1a* and *HR96-LBD-1b* alters detoxification gene expression.”, and we are space limited for an already long manuscript. We also don't see how the RNAi results (post-transcriptional) are related per se to the genome structure.

L334: IMPORTANT: it should be possible to infer the hierarchical order of regulation with your expression and mapping data.

We must admit that we do not understand why this comment is being made at this sentence in the Discussion, and we hope we are not missing the Reviewer's intent. However, we comment on the idea of inferring hierarchical gene expression above (see response to major point 2 of Reviewer 2).

L336: plant-produced aromatic compounds: such as? Please provide concrete examples!

As the reference reveals, there are many plant produced aromatic compounds metabolized by different DOGS, but we reformulated this sentence to be more precise: “a member of a detoxification gene family with broad substrate specificity against plant-produced mono- and polycyclic catecholic compounds”.

L337-338: this should be a separate sentence, while providing more points of discussion for those results.

The sentence was cut in two for clarity. The second sentence now reads : “Further supporting a role for *HR96-LBD-1a* and *HR96-LBD-1b* in the regulation of host plant use associated genes, many of the genes that responded to RNAi treatment also changed expression upon host shift from bean to tomato.”

L343: diverse acaricides such as?

We have given examples: “such as pyflubumide, abamectin, cyenopyrafen and fenpyroximate”.

L343-345: this information should be found in the Materials and Methods section.

We have removed this sentence from the Discussion, and added this detail and others about the strains used in the study to the Methods section “Mite strains and husbandry”, and also our response to Reviewer 3 point 1.

L348-353: could be streamlined. Also, it is not clear here to which work you are referring to. Snoeck et al.?

Yes, we are referring to Snoeck et al. The beginning of this sentence now reads “Collectively, these QTL findings...” and we cite Snoeck et al. again (which lengthens the sentences, not shortens them, but we do not see how these can be easily streamlined given the message).

L348: which cytochrome P450 reductase? This is a multi-gene family.

Cytochrome P450 reductase is a single copy gene in animals, including the spider mite, and in contrast to plants which have 2-3 paralogs.

L352: 'potentially the other acaricides to which the R strains is highly resistant': which other?

This is now more clearly explained in the Methods section “Mite strains and husbandry”.

L353: sorry, I don't understand this reasoning.

QTL mapping of acaricide resistance in a very closely related strain to the R strain revealed a QTL for CPR, suggesting that CYP mediated detoxification of tebufenpyrad is important for resistance to this compound (see Methods). This suggests a role for CYPs in the detoxification of this compound, but potentially also the many other compounds that the R strain is resistant to (CPR is required for CYP activity altogether). And, another QTL mapped to *HR96-LBD-1a* and *HR96-LBD-1b*, a regulator of CYP RNA levels as determined by our study (with the R strain haplotype leading to high upregulation, which could also increase detoxification potential). So, metabolic resistance to compounds to which the R strain is resistant is known or very likely to be CYP mediated, and the R strain locus for *HR96-LBD-1a* and *HR96-LBD-1b* is at a resistance QTL and the R haplotype massively upregulated some CYPs in *trans*.

L354-370: this section can be streamlined. Also, it seems largely speculative.

We have clarified some of this section, but added new information obtained in addressing comments of Reviewer 3.

L372-375: this is a very interesting result, but it should be discussed more in depth.

This now reads: “We also observed that a subset of genes that responded to RNAi knockdown of *HR96-LBD-1a* and *HR96-LBD-1b* differed between mites on bean versus tomato. This result suggests that signals from host plants, perhaps via plant specialized compounds, can modulate signaling by HR96-LBD proteins.”.

L376-382: content seems a bit repetitive and redundant.

We agree and have revised the last paragraph of the Discussion to be shorter and more direct.

Reviewer #3 (Remarks to the Author):

The manuscript by Ji et al. describe elegant experiments that lead to substantial results. eQTL mapping and consequential genetic experiments, exhibiting a beautiful reductionist logic, identify a major trans-regulator of detoxification genes in the spider mite. The significance is multi-faceted; (i) there is new insights into a chemical resistance in a major pest, (ii) here is a new and rare example of insecticide resistance mediated by a trans factor, (iii) that resistance relates to gene copy number variation and divergence between those copies, (iv) these copies encode a nuclear hormone receptor that has ligand binding but not DNA binding domains, (v) here's a molecularly defined example of epistatic interactions relevant to a newly evolved fitness trait (vi) and enticingly, could this help explain dramatic changes in transcriptomes and general vigour of arthropods that switch host plants?

The eQTL experiment cleverly exploited the haplodiploidy of the tiny mites to generate 458 recombinant inbred families. RNA-Seq from the families not only yielded the transcript abundance traits that were mapped but also the SNPs that allowed mapping (prior genome sequencing of the parental strains was used as a reference). One trans-eQTL then became the focus of the paper because it upregulated multiple detoxifying genes, including those associated with insecticide resistance. The QTL was validated in two ways. Firstly, by independently introgressing the region into a common background to generate Near Isogenic Lines, that were then transcriptionally profiled. Secondly, the trans QTL was then validated by RNAi against a candidate gene – a nuclear hormone receptor gene (assessed by further transcript profiling). A nice experiment that arose from an exploration of the data was a test for epistasis between trans eQTL and a cis eQTL. The methodology seems really solid and I can see no flaws.

The following suggestions are mere trinkets on christmas tree:

[Please note: A document with the changes made in response to all reviewers is provided with the changes shown with the track change feature. Please also note that we added fold change information to what was originally Supplementary Data 8, but is now Supplementary Data 11 in the resubmission. This makes it easier to compare fold changes across the experiments performed in the study.]

First, we appreciate the thoughtful and helpful comments.

1. All the variation considered existed between the two progenitor lines. We are told they differ in insecticide resistance profile, but the paper would be enriched by knowing more about the provenance of the lines. How many generations had they been in the lab? Were they collected off the same host plant? What host plant had they been reared on? Do they come from similar geographic regions? Should we think of these mites as panmictic?

In an effort to reduce the length of the manuscript for initial submission, we removed much of this information; we have now added it back (see also response to Reviewer 1, point 1). Additional information about the strains is now added at the Results section “Dense recombination in *T. urticae*.” (albeit very briefly), and more information about collection dates, hosts, acaricides to which the strains are resistant, and the respective classes (modes of action) are now provided in the Methods section “Mite strains and husbandry”. Additional collection information for the R and S strains, as well as 20 other inbred strains now included in the study (see response to Reviewer 3 point 5 below), is also provide in Supplementary Data 12 (which is new to this submission). With respect to being panmictic, please see response to Reviewer point 5 below.

2. Fig1a helped me understand the lines 92-97. The text (and my little-bit-of-knowledge-that-is-too-much)

didn't reveal that sons inherit recombined chromosomes (it now seems obvious that recombination occurs in the eggs that are 17n fertilized although pseudo-arrhenotoky does exist: so I am not recommending a change here just noting a path I went down). More importantly, I wonder whether the word 'samples' (192 and 199) is better replaced with something more precise (I used 'families' above).

We struggled with the nomenclature for the original submission, and based on the Reviewer's comments, which we welcome, it seems clear that our solution was not clear. One of several potential issues is that we refer to the F3 families as "populations", but collectively these 458 families (which are isogenic) are what we also called the eQTL mapping "population". Our use of samples to refer to the families appears to have further clouded understanding of the study.

We have adopted Reviewer 3's suggestion to use "families" instead of populations to describe the F3 cohorts. We also make it clear that the F3 families consist of genetically identical (isogenic) full siblings (nomenclature that we also think is easier to understand). Population is reserved to talk about the eQTL mapping population, and where we could use "families" instead of samples, if appropriate, we have done so. This involved changes to the main text, Fig. 1 ("Pop." To "Fam."), and the supplemental materials to be consistent (the changes are tracked).

Especially, please see the changes to the Results section "Dense recombination in *T. urticae*" and to the description of the experimental design in the Fig. 1 legend.

3. The b copy of the nuclear hormone gene HR96-LBD-1 gene differs between the resistant and susceptible strain by a radical amino acid substitution W309R. Is it worth inspecting this using alphafold structure predictions? Even if you look at the homologous region in the *Drosophila* version which is available through flybase would probably add more to supp figure 8.

Reviewer 1 also asked that we further investigate the potential impact of the W309R substitution. We used the alphafold-based program colabfold as well as sequence alignments to determine that the position of the W309R mutation is at the bottom of the ligand binding pocket and not in the dimerization domain (strikingly, it is predicted to be one of the key inward facing residues in the ligand-binding pocket itself). We added a sentence to the main text to state this observation (please see the last sentence of the second paragraph of Results section "Structural and allelic variation in HR96-LBD-1a and HR96-LBD-1b"). Further, we provide a detailed explanation and plots/alignments in a new supplementary figure (Supplementary Fig. 11 in the resubmission). A new methods section, "HR96-LBD-1a and HR96-LBD-1b alignments, homology modelling of HR96-LBD-1b, and alignment with known LBD structures", has been added to support this work (some text from the previous Methods section that described the HR96-LBD-1a and HR96-LBD-1b alignments is also now included in this Methods section in an effort to reduce the manuscript's length).

Quickly, this new analysis supports our original Discussion, as well as the one in this resubmission. Our hypothesis is that the R strain HR96-LBD-1b protein might have changed ligand specificity, or that the inward facing and radical W309R substitution might cause conformational changes in the ligand binding pocket to lead to constitutive activation (that is, the change makes it appear always ligand bound and activated). The change is not predicted to impact (directly) the LBD's dimerization or activation domains – this means that HR96-LBD-1b could still, potentially, dimerize with other NHRs with DNA-binding domains to impact detoxification gene expression.

Having said all this, we have not greatly expanded our text in the Results and Discussion as we do not want to introduce excessive conjecture. But, we appreciate the Reviewer's (and Reviewer 1's) prompting us to do the analysis, as what we do state is now on more solid footing.

4. Is it worth looking for common motifs in each gene that is regulated by transQTL 1? Would such analysis explain why some genes are more greatly upregulated?

This is a very reasonable idea, and in fact, yes, we tried this already (and we retried it in response to this Reviewer's question). For instance, we used with the 1000 bp upstream of genes for motif enrichment analysis for genes with *trans*-driven regulation from HS1 (AME v5.5.2; <https://meme-suite.org/meme/tools/ame>). We did not find significant enrichment of given motifs. However, given the phylogenetic distance of the invertebrate motif databased we used (based mostly on insects, but the most relevant one), we do not view this as surprising and we considered this a long shot to begin with (mites are chelicerates, and the distance to insects is > 450 million years).

5. It would be an even better paper if there was evidence that the trans-QTL showed signs of being recently selected. Could a junction spanning PCR trivially assess how common the CNV is at HR96-LBD-1? Are there signs of selective sweeps at the locus?

We believe that Reviewer 3 has picked up on something that is likely to be a common (*or maybe even very common*) reaction from readers of the manuscript. We agree that we should have done more in the original submission to investigate this, and we have done so in this resubmission. We think that our results clarify what might be a misunderstanding, and we also found a result with one strain that was not expected (but that may be relevant to understanding host plant interactions).

From our earlier studies, we published Illumina DNA-seq read data for 22 strains including the R and S strains. We aligned this existing read data for these 22 strains to the London genome sequence, which has one copy of HR96-LBD-1. From the read depth, we then estimated copy number variation (as normalized to the genome-wide background, which is expected to be mostly single copy, if a gene is duplicated it will have a normalized copy depth in aligned reads of ~2, if it is single copy, the normalized depth will be ~1, etc.).

For 21 of the 22 strains, the coverage depths were close to 2, suggesting that all these strains have two copies (coverage depths of about 1.7-1.9, including for the R and S strains, for which we know from the ultra-high quality PacBio assemblies that there are two copies; the slight deviations from 2 are not unexpected because genetic variants between strains can prevent the alignments of some reads). We actually noted this increased coverage originally for the R and S strains and that was one motivating factor to perform PacBio assemblies (we now note this explicitly in the text at the location where we discuss the rationale to perform the PacBio sequencing and assemblies, see Results section "Tandemly duplicated nuclear hormone receptor-96 (HR96) like genes at HS1.>").

Our new results allow us to infer that most *T. urticae* strains have two copies, and we strongly suspect that the single copy in the London genome sequence was an assembly error, but we do not know (the London genome was sequenced with the Sanger method which gives high quality reads, but they are low coverage because of cost, and assembly collapses at repeats can be common in such assemblies; this possibility is mentioned in the revised second to last paragraph of the Discussion).

One strain (C1N1d), however, had no read coverage, suggesting a deletion, which we confirmed by de novo assembly of this strain (a ~9 kb deletion removes both HR96-LBD-1a and 96-LBD-1b in this strain, as well as a flanking gene that is mentioned in a new Supplementary Figure, Supplementary Figure 8 in the resubmission, but that other gene is not relevant for the phenotypes studied in this work, see the legend). This strain is unique among the strains we examined (and among *T. urticae* populations at large) in that it is one of the only known host-race strains of *T. urticae* (it exists only on European honeysuckle

in the coastal dune ecosystem of the Netherlands). This is now mentioned in the Discussion (see also below).

Because the reviewer asked about CNVs and selective sweeps, and because we did the requested CNV analysis, an obvious question then becomes whether the variant for the W309R change is also found in other strains, especially ones with a history (or likely history) of acaricide exposure. The short read data for the strains (excluding R and S) made construction of the full sequences of the duplicate genes (which are of course repetitive) problematic. But, in the read alignments used for the CNV analysis, we did recover variants that gave rise to the W309R change in five strains other than the R strain (aligned Illumina reads at the respective position were extracted, and if multiple bases were present, targeted de novo assemblies were performed conditioned on the alleles, and fixed differences between *HR96-LBD-1a* and *HR96-LBD-1b* were then used to assess if the changes were in the a or b duplicate copies). Strikingly, the nucleotide differences giving rise to the W309R change were always in *HR96-LBD-1b*, and all the strains with the W309R change have known, or likely, acaricide exposure (including multi-acaricide resistant strain MAR-ABi, see the revised Discussion, second to last paragraph; strains collected from greenhouses, or indoors which usually come from greenhouses, that typically use chemical control). Moreover, two independent mutations gave rise to the W309R substitution, and were validated by PCR and Sanger sequencing in a subset of strains. This pattern is reminiscent of the independent origin of target-site mutations observed in *T. urticae* populations globally.

Our new findings are covered in the Results section as follows: (1) Beginning of the second paragraph of “Tandemly duplicated nuclear hormone receptor-96 (HR96) like genes at HS1”, and in (2) the new Results section “Structural and allelic variation in HR96-LBD-1a and 96-LBD-1b”. We discuss these findings, in the context of acaricide resistance and host plant use, in the revised Discussion (see paragraphs 5 and 7, track changes).

Supporting these analyses, two new Supplementary Data files were added (Supplementary Data 12 and 22 in the revised submission), two new Methods sections were added (“Characterization of HR96-LBD-1 copy number in a global collection of *T. urticae* strains” and “Allelic variation in HR96-LBD-1a and HR96-LBD-1b underlying the W309R change”), and three new Supplementary Figures were added (Supplementary Figures 8, 11 and 12). The respective sequence data have been deposited at NCBI (see revised Data Availability statement).

Finally, while the above analyses do, we believe, significantly add to our understanding of variation in HR96-LBD-1 genes among *T. urticae* strains, with 22 strains we are unable to perform the types of analyses that are needed to rigorously test for selective sweeps (such studies typically involve many tens to hundreds or even thousands of strains/individuals). Also, little is known about the demography of *T. urticae* strains, including the 22 we used (hence, the field just does not currently have the data to rigorously examine assumptions, like panmictic populations, that impact detection of selection).

While the analyses described above do constitute a significant addition to the manuscript, we remain circumspect in the Discussion in our treatment of the findings when appropriate, including noting that we still, of course, do not definitively know the causal variant.

6. Because time is needed to evolve sophisticated regulatory responses, induction may be more important for ‘plant chemical challenges’ than ‘pesticide resistance’. It is very exciting that you now associate transregulatory changes with the latter. But I feel that there is something more general that can be made of the former: what do these results say about host switching responses in arthropods more generally (anything to say to those studying whiteflies and aphids)?

As covered in our Discussion (original and revised), our results suggest that variation in a regulatory pathway that is likely evolved in the context of host plant interactions was co-opted to underlie the very high expression of detoxification genes that has now many times been associated with pesticide resistance in arthropods. It is probably a stretch to make definitive claims about host switching based on our findings. However, in the Discussion, we do now mention host breadth in the context of our findings with the C1N1d strain, which is an example of a rare host-race specialist strain of *T. urticae*, and that has also lost *HR96-LBD-1a* and *HR96-LBD-1b* entirely (but even here, we do need to be circumspect in what we can claim, and we have done so). Please see the tracked changes in the second to the last paragraph of the Discussion, in which we now also mention whiteflies briefly (the well-established findings for *Bemisia tabaci*).

Reviewers' Comments:

Reviewer #1:

Remarks to the Author:

The authors have carefully responded to my criticisms and suggestions of the previous version. The most important suggestion was followed and the results of the further analysis strengthen the paper. The inability to follow other suggestions was convincingly justified. I find this to be the case in the responses to the other reviewers' comments as well. I am entirely satisfied with the result.

Reviewer #2:

Remarks to the Author:

The authors have nicely addressed all major concerns as relevant to their study.

One last item: could the authors please better highlight for Fig. 4 the connection to the four other selected genes explicitly highlighted in Fig. 4A that is UGT204B1/B2; CCE58; UGT201B12; CYP392EnP, which could be provided in the figure legend ?

While you refer to Fig. 3 it's not that obvious, especially for UGT201B12. This would also require a referral within the main text, and a brief explanation.

Reviewer #3:

Remarks to the Author:

Thankyou for the thorough response to the first round of reviews. Congratulations on some excellent science.

“Reviewer #2 (Remarks to the Author):

The authors have nicely addressed all major concerns as relevant to their study.

One last item: could the authors please better highlight for Fig. 4 the connection to the four other selected genes explicitly highlighted in Fig. 4A that is UGT204B1/B2; CCE58; UGT201B12; CYP392EnP, which could be provided in the figure legend ?

The respective sentence now reads: “Select detoxification genes (filled circles, legend, bottom right) with large HS1 *trans*-driven fold changes (*CYP392A12*, *UGT204B1*, *UGT204B2*, *CCE58*, and *CYP392EnP*, see Fig. 3c) are labeled, with *CYP392A12* in bold text).” This now makes it clear what is being referenced in Figure 3 (these genes are explicitly labelled in Fig. 3c).

While you refer to Fig. 3 it's not that obvious, especially for UGT201B12. This would also require a referal within the main text, and a brief explanation.

It was an oversight to label *UGT201B12* in Fig. 4a (we only meant to label genes that were also labelled in Fig. 3c). We have removed the *UGT201B12* label from Fig. 4a. We certainly see why this was confusing, and this resolves the source of confusion (with this fix, no changes are needed to the main text).